# DEQuify your force field: More efficient simulations using deep equilibrium models

## Abstract

Machine learning force fields show great promise in enabling more accurate force fields than manually derived ones for molecular dynamics simulations. State-of-the-art approaches for ML force fields stack many equivariant graph neural network layers, resulting in long inference times and high memory costs. This work aims to improve these two aspects while simultaneously reaching higher accuracy. Our key observation is that successive states in molecular dynamics simulations are extremely similar, but typical architectures treat each step independently, disregarding this information. We show how deep equilibrium models (DEQs) can exploit this temporal continuity by recycling intermediate neural network features from previous time steps. Specifically, we turn a state-of-the-art force field architecture into a DEQ, enabling us to improve both accuracy and speed by $10\% - 20\%$ on the MD17, MD22, and OC20 200k datasets. Compared to conventional approaches, DEQs are also naturally more memory efficient, facilitating the training of more expressive models on larger systems given limited GPU memory resources.

## 1 Introduction

With increasingly more available compute, molecular dynamics (MD) simulations emerged as an integral tool for studying the behaviour of molecules to develop a mechanistic understanding of a large class of processes in drug discovery and molecular biology Lin & MacKerell (2019); Hollingsworth & Dror (2018); Sinha et al. (2022); Durrant & McCammon (2011). The backbone of an MD simulation is a force field, which predicts the forces acting on each of the atoms in a molecule, given the current atom positions. These forces are then used to integrate the equations of motion numerically by multiplying the forces with a small time step $dt$ to obtain velocities, which in turn is used to update the atom's positions. Traditionally, force fields were designed by hand to capture known physical effects such as covalent bonds, electrostatics, and van der Waals forces Weiner & Kollman (1981); Pearlman et al. (1995). These hand-crafted force fields are compact and fast but lack the expressivity to capture more complex quantum mechanical many-body interactions. Alternatively, force fields can be calculated from highly accurate but costly quantum mechanical calculations, so-called ab-initio methods. Therefore, a new approach has gained traction over recent years: Training an expressive machine learning model on data from expensive ab-initio methods. This results in models at near-quantum chemical accuracy at only a fraction of the cost.

Some early works on machine learning force fields used local atom environment descriptors in combination with linear regression Thompson et al. (2015); Shapeev (2016), Gaussian processes Bartók et al. (2010), and feed-forward neural networks Behler & Parrinello (2007). The pioneering work SchNet Schütt et al. (2017) used a rotation invariant graph neural network to predict energies, forces and other properties. This was later improved by the use of equivariant neural networks that model angular dependencies more directly, such as Cormorant Anderson et al. (2019), DimeNet Gasteiger et al. (2020), PaiNN Schütt et al. (2021), GemNet Gasteiger et al. (2021), SphereNet Liu et al. (2022), and NequIP Batzner et al. (2022). Recent models have further improved the expressivity and scalability of equivariant models. Equiformer introduces an attention mechanism Liao & Smidt (2023), Allegro focuses on edge features with non-growing receptive fields Musaelian et al. (2023), MACE introduces an efficient mechanism to calculate many-body interactions with high-order tensor polynomials Batatia et al. (2022), eSCN improves the scaling of Clebsch-Gordan products involved in equivariant convolutions Passaro & Zitnick (2023), and ViSNet

Wang et al. (2022) and QuinNet Wang et al. (2024) derive ways to incorporate four and five body terms much more efficiently. CHGNet Deng et al. (2023) incorporates magmoms for extra physical supervision. VisNet-LSRM Li et al. (2023) and 4G-HDNNP Ko et al. (2021) focus on modelling long-range and non-local effects. The latter is the most related to our current works, as it employs a physically motivated charge equilibration scheme to construct input features to their neural network. All the works mentioned above are neural networks built on message-passing schemes that respect permutation, translation, rotation and optionally inversion symmetries, leading to much-improved sample efficiency.

Despite these advancements, the computational cost of accurate predictions remains a significant challenge. Even though machine learning force fields are much faster than quantum chemical calculations, simulating systems with these models is still expensive. This is because, over the course of a full MD simulation, the force field has to be called millions to billions of times Hollingsworth & Dror (2018). Even for relatively cheap machine-learning force fields, this incurs a high cost. Additionally, equivariant networks are memory-consuming due to the nature of the equivariant message-passing operation Passaro & Zitnick (2023), complicating the training of expressive models on large systems. Thus, compute-efficient models are of great interest.

This work addresses the challenge of fast inference speed, low memory footprint, and high accuracy by introducing the Deep Equilibrium Networks (DEQ) formalism Bai et al. (2019) to equivariant architectures. DEQs replace the typical deep stack of layers with fewer layers and a fixed-point solver (see section 2.3), which results in more expressive models given a fixed parameter count. Memory efficient gradient computation is enabled by the implicit function theorem, with constant cost independent of number of function calls in the solver. Crucially, we exploit the temporal correlation between successive states in MD simulations to reduce the number of solver steps to find the fixed-point, gaining a speedup. By reusing the fixed-point from the previous MD state as an initial guess to warm start the following fixed-point solver iteration, we effectively "share compute" between time steps (see figure 1).

We implement our method by adapting the EquiformerV2 architecture Liao & Smidt (2023); Liao et al. (2024) since at the time of writing it holds the top spots on the Open Catalyst Project leaderboard. [1] In principle however, the methodology is compatible with other similar force field architectures Gasteiger et al. (2021).

Our results show that, compared to the original EquiformerV2, DEQuiformer achieves (1) 10-20% faster inference and equally or higher accuracy on the MD17/MD22 dataset, (2) significantly improved accuracy for the OC20 200k dataset, (3) all at reduced training memory cost and (4) with 2-5x fewer model parameters.

We summarize our contributions as follows:

1. This is the first work to propose the use of DEQ with equivariant networks, and its application to ML force fields

2. We demonstrate that it is possible to exploit the temporal continuity in molecular dynamics, by reusing sequential fixed-points in DEQs, on common datasets and practical simulations

3. Show that implicit models for ML force fields can improve upon speed, accuracy, training memory, and parameter efficiency compared to their explicit counterparts

## 2 PRELIMINARIES

State-of-the-art ML force fields like EquiformerV2 belong to the class of equivariant graph neural networks (GNN) Batatia et al. (2022); Musaelian et al. (2023); Liao et al. (2024); Batzner et al. (2022). The central shared feature is the stacking of equivariant message passing layers, typically between five Batzner et al. (2022) and twenty Liao et al. (2024). 3D rotational and translational equivariance is achieved by building on irreducible representations and spherical harmonics, improving data efficiency Batzner et al. (2022). We briefly introduce the idea of these equivariant GNNs.

---

[1]https://opencatalystproject.org/leaderboard.html

## 2.1 EQUIVARIANT GRAPH NEURAL NETWORKS

Molecules can be seen as graphs living in Euclidean 3-dimensional space. As such, they adhere to a set of known symmetries that a equivariant models exploit. Such symmetry exploiting networks have emerged as the SOTA for molecular data Passaro & Zitnick (2023); Musaelian et al. (2023); Batatia et al. (2022); Liao et al. (2024); Thomas et al. (2018). For example, the energy of a molecule does not change if we rotate it or if we permute the index of two atoms; therefore, we call the energy rotation and permutation invariant. Forces, on the other hand, rotate along with the molecule, which we call rotation equivariant.

A Graph Neural Network (GNN) takes in a graph $\mathcal{G}$ and maps it to a target space in a permutation equivariant way. If the graph is embedded in 3d space as molecules are, we use $O(3)-$Equivariant graph neural networks, which are equivariant to translations, rotations and optionally inversions. In these networks, node features $h_t$ of node $t$, are concatenations of irreducible representations (irreps) $h_t^l \in \mathbb{R}^{2l+1}$, organized by their degree $l$ (we omit an additional channel dimension for simplicity). Irreps transform under rotation $R$ as

$$h^l(R \cdot (r_1, ..., r_N)) = D_l(R) \cdot h^l(r_1, ..., r_N) \tag{1}$$

where $r_1, ..., r_N$ are the coordinates of the atoms, and $D_l(R) \in \mathbb{R}^{(2l+1)\times(2l+1)}$ is the Wigner-D matrix. Intuitively, higher-degree features rotate faster with rotation of the input features. $l = 0$ features are rotation invariant scalars, and $l = 1$ are ordinary vectors. A vector $r \in \mathbb{R}^3$ can be mapped to an $l$ graded feature using the spherical harmonics $Y_l(r/||r||) \in \mathbb{R}^{2l+1}$.

Two irreps $f^{l_1}$ and $g^{l_2}$ of different degrees interact using the Clebsch-Gordan tensor product Thomas et al. (2018)

$$h_{m_3}^{l_3} = \left( f_{m_1}^{l_1} \otimes_{l_1,l_2}^{l_3} g_{m_2}^{l_2} \right)_{m_3} = \sum_{m_1=-l_1}^{l_1} \sum_{m_2=-l_2}^{l_2} C_{(l_1,m_1),(l_2,m_2)}^{(l_3,m_3)} f_{m_1}^{l_1} g_{m_2}^{l_2} \tag{2}$$

where we index the elements within the $2l + 1$ dimensional tensor by $m$, and $C_{(l_1,m_1),(l_2,m_2)}^{(l_3,m_3)}$ are the Clebsch-Gordan coefficients. Every combination of $l_1, l_2, l_3$ is called a path, and every path is weighted individually by $w_{l_1,l_2,l_3}(\cdot)$. The weight itself is predicted by a neural network, conditioned on rotation invariant features like the distance $||r_{ts}||$.

An equivariant GNN builts on top of equation 2 to define an equivariant message passing scheme: Given a target node $h_t$ and a source node $h_s$ with a relative coordinate vector $r_{ts}$, an equivariant GNN sends a message from the source to the target using

$$v_{ts}^{l_3} = v^{l_3}(h_t, h_s, r_{ts}) = \sum_{l_1,l_2} w_{l_1,l_2,l_3}(||r_{ts}||) \left( f^{l_1}(h_t, h_s) \otimes_{l_1,l_2}^{l_3} Y^{l_2}(r_{ts}/||r_{ts}||) \right) \tag{3}$$

where $f(h_t, h_s)$ is a function of both target and source node features; in EquiformerV2, it is simply the concatenation operation. Instead of using equation 3 directly, EquiformerV2 relies on eSCN convolutions, which calculates basically the same expression but in a more efficient way; please refer to Passaro & Zitnick (2023) and Liao et al. (2024) for details.

## 2.2 EQUIFORMERV2

EquiformerV2 Liao et al. (2024) in particular is a graph transformer, where each message passing layer is an equivariant transformer block. To initialize the node features, the embedding block first encodes the input molecule, based the atom numbers $\mathbf{z}$ and positions $\mathbf{r}$.

$$\mathbf{h}_i^{(0)} = \text{Embed}(x_i) = \text{Embed}(\mathbf{z}_i, \{\mathbf{r}_{ij}\}_{j \in \mathcal{N}(i)}) \tag{4}$$

The $L$ transformer layers then perform repeated attention-weighted message passing to update the node features based on nodes in the neighbourhood.

$$\mathbf{h}_i^{(l+1)} = f_\theta^{(l)} \left( \mathbf{h}_i^{(l)}, \{\mathbf{h}_j^{(l)}, \mathbf{r}_{ij}\}_{j \in \mathcal{N}(i)} \right) \tag{5}$$

After several transformer blocks update the node features, they are passed to two separate output heads for the final force and energy predictions. The total energy of the molecule is just the sum of

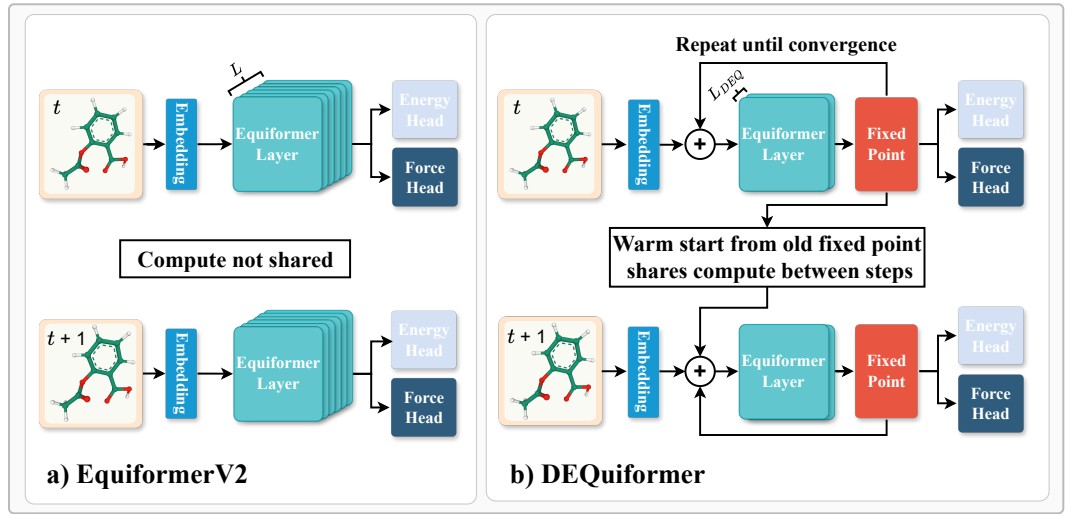

Figure 1: Comparison of the EquiformerV2 and DEQuiformer architectures. While the Equiformer model considers every input state independently, the DEQuiformer exploits the temporal continuity between input states to share compute. This works because neighbouring time steps in an MD simulation are highly similar. Therefore, we drastically reduce the required compute by reusing the fixed-point from the previous step.

the energies of the individual nodes.

$$E = \sum_i Out^{\text{scalar}}\left(\mathbf{h}_i^{(L)}\right) \tag{6}$$

$$\mathbf{F}_i = Out^{\text{vector}}\left(\mathbf{h}_i^{(L)}, \mathbf{z}_i, \mathbf{r}_i j\right)_{j \in \mathcal{N}(i)} \tag{7}$$

We provide more details on equivariant GNNs and EquiformerV2 in section A.1.1.

## 2.3 DEEP EQUILIBRIUM NETWORKS

**Implicit models**    Most machine learning models, like EquiformerV2, are explicit. Explicit models are defined by mapping the input to an output via a fixed computational graph. For example, most traditional deep learning models stack a fixed number $L$ of layers such that the depth of the model does not change during runtime. In contrast, implicit models, which have recently surged in popularity, define their output as the solution to a learned dynamical system. Two prominent examples of implicit models are Neural ODEs and Deep Equilibrium Models (DEQ).

DEQs compute their output as the fixed-point of an input-conditioned mapping. In contrast to explicit models, this cannot be expressed as a fixed computational graph. We can use different root solvers, each yielding different performance characteristics. If we roll out the solver trajectory, the iterations take the form of weight-tied layers. Therefore, one often says these models have "continuous layers" or "infinite depth". One of the main selling points of implicit models is that they require constant memory with respect to their "depth" during training, independent of the solver used. This starkly contrasts explicit models, where the memory complexity grows linearly with each layer.

**Implicit layer**    Deep Equilibrium models drastically reduce the model size by replacing the deep stack of layers with just one or two layers and a fixed-point solver. In early work Bai et al. (2018), it was first shown that a stack of $L$ residual layers yields very competitive results even if all layers are weight-tied. In Bai et al. (2019), the authors showed empirically that the network converges to a fixed-point in the limit of infinite depth $L \to \infty$. To formalize this, consider a function $f_\theta$, usually a small neural network. Given some input $x$, repeated passes through $f_\theta$ updates the features $\mathbf{h}^s$

$$\mathbf{h}^{i+1} = f_\theta(\mathbf{h}^s, x) \tag{8}$$

until the features converge to a fixed-point $\mathbf{h}^* = f_\theta(\mathbf{h}^*, x)$. The "fixed-point" or "equilibrium point" is then considered the output of the fixed-point layer. This replaces the intermediate features $\mathbf{h}^{(l)}$ after $l$ layers with a fixed-point estimate of the features $\mathbf{h}^s$. Notice that the neural network layer $f_\theta$ has to take in the input $x$, in addition to the current features $\mathbf{h}^s$, at every pass, which is called the input injection. This setup reduces the model from many layers to just a few. Naively passing $\mathbf{h}$ through the NN layer $f_\theta$ many times renders this approach slow, though. Instead, we search for the fixed-point directly by using a root-solving algorithm like Anderson acceleration Anderson (1965) or Broyden's method Broyden (1965), which computes more sophisticated updates of $\mathbf{h}$ to reduce the number of passes until the fixed-point is reached.

**Memory efficient gradient**   Backpropagating through this solver trajectory would incur a prohibitive memory cost. Fortunately, a unique feature of DEQs is that the gradient can be computed by the Implicit Function Theorem (IFT) Bai et al. (2019):

$$\frac{\partial L}{\partial \theta} = \frac{\partial L}{\partial \mathbf{h}^*} \left( 1 - \frac{\partial f_\theta}{\partial \mathbf{h}^*} \right)^{-1} \frac{\partial f_\theta (\mathbf{h}^*, x)}{\partial \theta} \tag{9}$$

Using IFT, the forward passes is performed without tracking gradients, i.e. without storing the layer activations. Thus, the memory cost during training, usually dominated by the gradient, becomes independent of the DEQ's "depth."

With the Implicit Function Theorem (IFT) the gradient is computed by solving a second fixed-point system, for which we again use a root solver Bai et al. (2019).

$$\mathbf{g}^* = \mathbf{g}^* \frac{\partial f}{\partial \mathbf{h}^*} + \frac{dL}{d\mathbf{h}^*} \tag{10}$$

Computing the gradient via IFT reduces the memory requirements during training, at the cost of extra time to solve equation 10.

Many DEQ works circumvent solving equation 10 by the so-called 1-step gradient approximation Fung et al. (2022) . Crucially we found that the 1-step gradient led to suboptimal accuracy in combination with EquiformerV2. Improving the stability of DEQ training and its gradients is an ongoing open problem that could improve the extended training times. We discuss this in more detail in section A.1.2.

## 3   METHOD: DEQUIFORMER

The central step to "DEQuify" EquiformerV2 is to replace the $L$ Equiformer layers with a fixed-point solver over $L_{DEQ} \ll L$ Equiformer layers, as shown figure 1. We now go through the steps necessary to make this work in practice.

**Embedding block for input injection**   Equiformer initializes the node features via an embedding block emb $=$ Embed $(x)$. Using the embedding to initialize the initial fixed-point estimate $\mathbf{h}^0$ however would stop gradients to flow to the encoder, since the gradient calculated via IFT is independent of the solver trajectory. Instead, we use the embedding block's output as the input injection, by adding the embedding to the fixed-point estimate $\mathbf{h}^s$ at every solver step before passing it through the layer $g_\theta$

$$f_\theta(\mathbf{h}^s, x) = g_\theta \left( (\mathbf{h}^s + \text{emb}) \frac{\|\text{emb}\|}{\|\mathbf{h}^s + \text{emb}\|} \right) \tag{11}$$

To prevent the norm of the features to grow with depth, we rescale the vector 2-norm $\|\cdot\|$ to be the same as before the addition. Following Bai et al. (2019), the node features are initialized as all zeros $\mathbf{h}^0 = 0$.

**Output head**   Like the embedding block, the output block itself remains unchanged. EquiformerV2 predicts the forces and energy via separate output heads, by acting on the node features after $L$ layers $\mathbf{h}^{(L)}$. The node features are instead replaced by the fixed-point estimate of the node features $\mathbf{h}^*$ from the root solver, which we pass as input to the output heads.

**Recurrent dropout**   Dropout is a widely used regularization that tends to hurt DEQ performance. This is because dropout samples a new mask for each pass through the implicit layer, which hinders finding a fixed-point Bai et al. (2019). EquiformerV2 uses two types of dropout, alpha dropout (acting on nodes) and path dropout, also known as stochastic depth (acting on edges). For DEQuiformer we instead use recurrent dropout, which applies the same mask at each step of the fixed-point solver, but a different mask for each sample Bai et al. (2019); Gal & Ghahramani (2016). While recurrent path dropout improved the generalisation of DEQuiformer, adding alpha dropout, either the recurrent or regular version, reduced the accuracy in DEQuiformer. We hypothesise this is because dropout is designed to reduce overfitting in large models, to which DEQs are less prone to due to their fewer parameters. We subsequently remove alpha dropout from DEQuiformer but keep alpha dropout in EquiformerV2.

**Training stability with fixed-point correction loss**   Without further regularization, DEQs may become unstable over the course of training, noticeable by increasing number of root solver steps Bai et al. (2022; 2021); Geng & Kolter (2023). A simple yet effective remedy is the *sparse fixed-point correction* regularization loss Bai et al. (2022). Given a fixed-point solver trajectory $\mathbf{h}^0, \cdots, \mathbf{h}^s, \cdots, \mathbf{h}^*$ we pick some fixed-point estimates $\mathbf{h}^s, s \in \mathcal{I}$ and add their gradient as if they were the final fixed-point estimate. We follow Bai et al. (2022) and uniformly pick three $\mathbf{h}^s$ along the solver trajectory.

**Fast inference via fixed-point reuse**   Our main observation is that consecutive time steps in a molecular dynamics simulation are highly similar; thus, their fixed-points should also be similar. At inference time, the number of solver steps can therefore be significantly reduced by initializing the fixed-point estimate not from all zeros but the fixed-point of the previous time step: $\mathbf{h}^0_{t+1} = \mathbf{h}^*_t$. The same idea was successfully demonstrated before in the context of optical flow prediction from videos Bai et al. (2022).

**Accuracy-compute tradeoff in the root solver**   To quickly reach a fixed-point we use Anderson acceleration as a root solver, since it is faster than naive fixed-point iteration while being more stable than Broyden's method. During training, we require low fixed-point errors to ensure that gradients can be calculated with the IFT. However, we can trade off performance and time during inference by relaxing the error threshold for the root solver Bai et al. (2022). With the right threshold, this significantly speeds up inference while only marginally affecting performance. For simplicity, we adhere to the settings of Bai et al. (2022). During training we stop after the absolute fixed-point error falls below a relative threshold $|f_\theta(\mathbf{h}^s) - \mathbf{h}^s|/\|\mathbf{h}^s\| < \epsilon_{train} = 10^{-2}$ or a maximum of 40 steps is reached. During inference, we compute the first fixed-point at the same tight tolerance $\epsilon_{test} = \epsilon_{train}$, but then relax the threshold for the following time steps to $\epsilon_{test}^{FPreuse} = 10^{-1}$. Relaxing the tolerance further reduces the number of forward steps and thus inference time, without sacrificing accuracy, as we show later.

## 4  EXPERIMENTS

In our experiments, we are focusing on the direct comparison between EquiformerV2 and it's DEQ-variant DEQuiformer on the most common MD dataset. In particular, we compare a 1 and 2-layer DEQuiformer with a 1, 4, and 8-layer EquiformerV2 on the MD17/MD22 dataset and up to 14 layers on the OC20 200k dataset. Our goal is to experimentally demonstrate the following points:

1. DEQs work with equivariant networks, with $l-$graded features and converge in a stable manner

2. Reusing fixed-points speeds up DEQuiformer, pushing the accuracy-speed Pareto front

3. DEQs scales to large datasets like OC20 200k, significantly improving peak accuracy

4. While using much fewer model parameters

Due to computational constraints, we use slightly smaller versions of EquiformerV2 and only the smaller 200k data split for OC20. We also reduce the model size for the much smaller MD17/MD22 datasets. For details, please refer to section A.2 in the appendix.

**MD17**   MD17 contains trajectories of molecular dynamics simulations of eight small molecules with 9 to 21 atoms. For each molecule, there are between 100,000 to 1,000,000 data points. Following Equiformer, a random subset of 950 data points is used for training, 50 for validation, and test on all the remaining samples.

Like EquiformerV2, we use the original MD17 instead of the revised MD17 dataset Christensen & von Lilienfeld (2020), which contains more accurate forces. The data points in rMD17 are not from sequential MD timesteps, which makes rMD17 unsuitable for benchmarking with fixed-point reuse. Since we are only interested in the direct comparison with EquiformerV2, using the lower-quality MD17 dataset does not favor one model over the other.

**MD22**   The MD22 dataset extends MD17 by seven larger molecules with 42 to 370 atoms Chmiela et al. (2023). We use the same datasplit for MD22 as for MD17. The double-walled nanotube is much larger than the other systems, causing the 8-layer EquiformerV2 to run out of memory on our compute setup. We, therefore, excluded the nanotube from the benchmark. However, this shows one of the strengths of DEQs, as it allows us to train expressive models even on large systems.

**OC20**   The Open Catalyst Project (OC20) offers a much larger dataset, containing 1.3 million molecular relaxations from 260 million DFT calculations. We restrict ourselves to the structure to energy and forces (S2EF) 200k split. While effective for evaluating the accuracy of our approach, the speedup with fixed-point reuse cannot be tested, since the samples are not temporally ordered and no time information is available.

**Evaluation**   The error is computed as the mean average error (MAE) over the test set. The energies and forces are in units of kcal/mol and kcal/mol/Å. Time is measured as the forward pass on an AMD MI100, averaged over the test set.

To aggregate the results over all molecules in MD17/MD22, we use minmax normalization per molecule, and report the mean and standard deviation of the mean. We describe the procedure in section A.2.

**Training**   Following previous work, we train separate models for each molecule. For MD17/MD22, we use a smaller model than the original EquiformerV2, which focuses on larger splits of OC20 (2M and above). For OC20 200k we use the default hyperparameters in the EquiformerV2 repository. For MD17/MD22 each model is trained on a single AMD MI100 GPU with 32GB GPU RAM for 500 epochs, which takes 10 to 30 hours. For OC20 200k training takes about 20 to 60 hours for three epochs. For simplicity, we use the same training hyperparameters from EquiformerV1 and EquiformerV2, without optimizing them for DEQuiformer specifically. A detailed discussion can be found in section A.2.

### 4.1   RESULTS

**Training dynamics**   Our first question is whether or not our DEQuiformer converges to a fixed-point. Since no prior work has combined DEQs with a rotation equivariant architecture, this is not at all obvious. To answer this question, we look at the relative fixed-point error on the Aspirin molecule as a function of the fixed-point solver steps at different steps in training; see figure 2a. We see that the fixed-point error decreases with the number of solver steps, as expected. The fixed-point iteration is stable over the training, even slightly improving, resulting in slightly faster convergence later in training.
Additionally, we look at the loss curves of aspirin as an example, figure 2b and figure 2c. We see that the training stability of DEQuiformer is similar to that of EquiformerV2. The same holds for OC20 200k; see figure 5a in the supplementary material. We can also see that our DEQuiformer has both better training and better test errors, indicating that the improvements are not just due to reduced overfitting from a lower parameter count.

**Speedup on sequential MD data**   Our second question is if DEQuiformer provides a speedup on the molecular dynamics data across MD17/MD22. Indeed, DEQuiformer achieves consistently faster inference speeds at better accuracy than EquiformerV2, see figure 3a. Focusing on a 1-layer DEQuiformer vs. a 4-layer EquiformerV2, we measure an average inference time improvement of 19

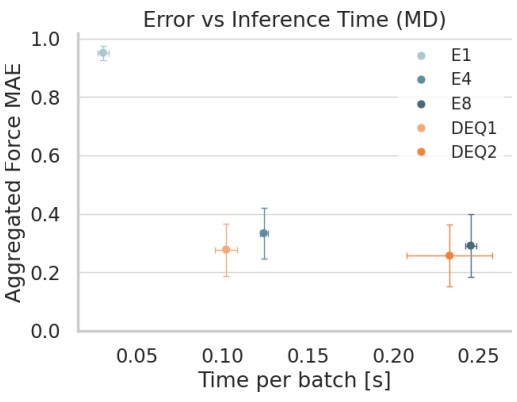

(a) DEQuiformer converges to a fixed-point and is stable over training.

(b) DEQuiformer trains faster, achieving lower train error.

(c) Lower train error translates to lower test error in DEQuiformer.

Figure 2: DEQuiformer enjoys stable training dynamics, reaching lower train and test error.

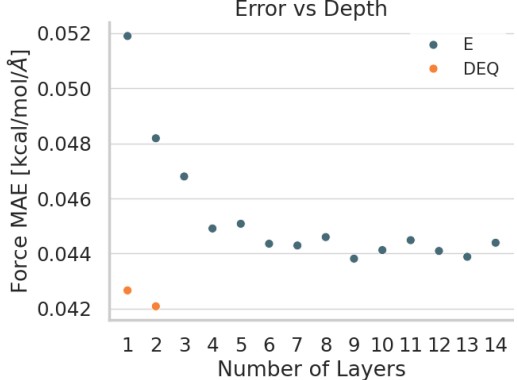

(a) Results for MD17/MD22: DEQuiformer is faster than Equiformer during inference at the same or better accuracy.

(b) Results for OC20 200k: DEQuiformer outperforms EquiformerV2 despite using much fewer parameters. Even with 14 layers (maxing out our memory), DEQuiformer still performs much better, indicating that DEQs are more data efficient for force fields.

Figure 3: Results on MD17/22 and OC20 200k: DEQuiformer is faster and more accurate than EquiformerV2 while also using much fewer parameters.

%, at 15 % better accuracy. The error bars are computed according to equation 20. A full breakdown of the force test errors and inference time can be found in table 1. Energy predictions are listed in the supplementary material under table 4, although note that there was a significantly higher weight on the force loss.

The speedup is possible through fixed-point reuse. We examine the impact of this in more detail: In figure 4b, we plot the number of solver steps needed to find the fixed-point in the test set on Aspirin with and without reusing the fixed-point from the previous time step. While DEQuiformer takes about 5-6 steps to find the fixed-point when starting from $\mathbf{h}^0 = 0$; this gets reduced to 3 steps if we warm-start the solver, supporting our claim that we "share compute" between successive time steps.

**Accuracy on larger dataset** Since MD17/MD22 is a comparatively small dataset, where more expressive models often may not perform better Liao et al. (2024), we validate the DEQ approach on OC20 200k. In figure 3b, we plot the force error over the number of layers, using a maximum of 14 layers for EquiformerV2, the maximum our GPU memory could support. DEQuiformer reaches significantly better accuracy than EquiformerV2 while using far fewer parameters. Interestingly, EquiformerV2 seems not to benefit much from an increase in depth after a certain point, such that a one- or two-layer DEQuiformer is outperforming even an 14-layer EquiformerV2. Quantitative results are in table 2b.

| MD17 | Ethanol | | Malonaldehyde | | Benzene | | Uracil | | Toluene | | Salicylic acid | | Naphthalene | | Aspirin | |
|---|---|---|---|---|---|---|---|---|---|---|---|---|---|---|---|---|
| | Force | Time | Force | Time | Force | Time | Force | Time | Force | Time | Force | Time | Force | Time | Force | Time |
| EquiformerV2 (1 layer) | 0.29 | 0.04 | 0.43 | 0.05 | 0.21 | 0.02 | 0.39 | 0.03 | 0.17 | 0.04 | 0.42 | 0.03 | 0.15 | 0.03 | 0.53 | 0.04 |
| EquiformerV2 (4 layers) | **0.22** | 0.12 | 0.31 | 0.15 | **0.20** | 0.13 | **0.30** | 0.10 | **0.13** | 0.12 | 0.33 | 0.12 | **0.12** | 0.13 | 0.42 | 0.11 |
| EquiformerV2 (8 layers) | 0.22 | 0.23 | 0.32 | 0.25 | **0.18** | 0.24 | **0.30** | 0.25 | **0.13** | 0.25 | 0.34 | **0.25** | 0.12 | 0.25 | 0.42 | 0.25 |
| DEQ (1 layer) | 0.22 | **0.09** | 0.31 | **0.09** | 0.22 | 0.13 | 0.35 | **0.09** | 0.15 | **0.09** | **0.31** | **0.06** | 0.15 | **0.09** | **0.40** | **0.09** |
| DEQ (2 layers) | **0.21** | **0.18** | **0.31** | **0.19** | 0.21 | **0.18** | 0.30 | **0.20** | 0.14 | **0.20** | **0.27** | 0.53 | **0.11** | **0.18** | **0.39** | **0.17** |

| MD22 | Ac-Ala3-NHMe | | DHA | | AT-AT | | Stachyose | | AT-AT-CG-CG | | Buckyball catcher | | # Weights |
|---|---|---|---|---|---|---|---|---|---|---|---|---|---|
| | Force | Time | Force | Time | Force | Time | Force | Time | Force | Time | Force | Time | |
| EquiformerV2 (1 layer) | 0.70 | 0.03 | 0.28 | 0.05 | 0.40 | 0.03 | 0.35 | 0.04 | 0.33 | 0.04 | 0.15 | 0.05 | 670k |
| EquiformerV2 (4 layers) | 0.32 | 0.13 | 0.21 | **0.11** | **0.29** | **0.11** | 0.23 | 0.10 | 0.23 | 0.12 | **0.09** | 0.12 | 1.7M |
| EquiformerV2 (8 layers) | 0.31 | 0.25 | 0.21 | **0.25** | 0.29 | **0.25** | 0.22 | 0.25 | 0.20 | 0.23 | **0.08** | 0.25 | 3M |
| DEQ (1 layer) | **0.32** | **0.10** | **0.19** | 0.16 | 0.30 | 0.15 | **0.22** | **0.09** | **0.21** | **0.10** | 0.14 | **0.09** | **670k** |
| DEQ (2 layers) | **0.29** | **0.19** | **0.20** | 0.33 | **0.27** | 0.28 | **0.20** | **0.19** | **0.18** | **0.20** | 0.15 | **0.25** | 1M |

Table 1: **Accuracy and speed on MD17 and MD22.** DEQuiformer is faster at comparable accuracy. Force MAE is in units of kcal/mol/Å. Time is measured as the average seconds per batch on a AMD MI100 GPU. Lower is better. We highlight the lowest error and time per batch comparing EquiformerV2 (4 layers) to DEQuiformer (1 layer), and EquiformerV2 (8 layers) to DEQuiformer (2 layers), since they respectively require a comparable amount of inference time.

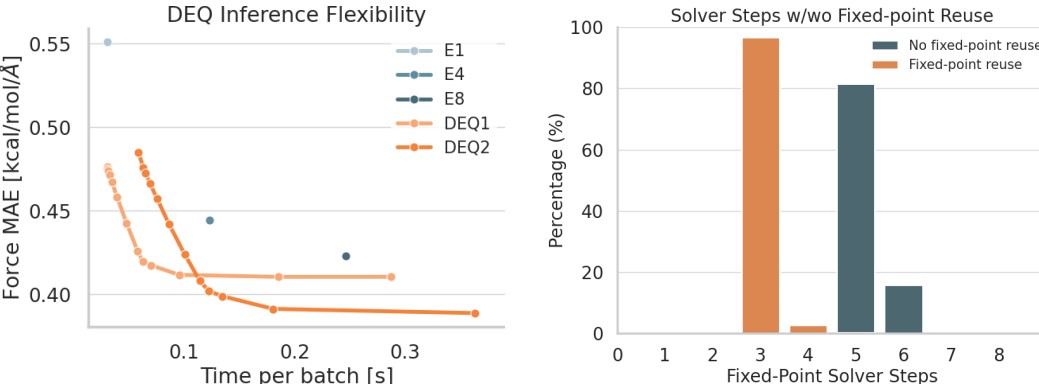

(a) Compute-accuracy-tradeoff at inference time. DE-Quiformer is remarkably robust to its fixed-point error up to a threshold of about $10^{-1}$, where the error starts to rapidly increase. As expected, higher fixed-point tolerances lead to faster inference speed.

(b) Reusing the fixed-point significantly reduces the number of solver steps in DEQuiformer to enable a speedup. Percentage denotes relative number of samples in the test set that required a given number of solver steps.

Figure 4: Examining DEQuiformers fixed-point behaviour.

**Trading off accuracy for speed**   A unique feature of DEQs is that we can trade off accuracy for extra speed post-training by loosening the fixed-point error threshold. The looser this threshold, the faster the model, since the fixed-point solver terminates earlier. We are examining how sensitive the force error is with respect to this fixed-point error tolerance. We calculate the validation error and time per batch for different solver tolerances on a logarithmic scale, with the Aspirin molecule as an example. The results are plotted in figure figure 4a. As expected, looser thresholds lead to faster inference time but higher force errors. Remarkably, the model's predictions seem robust until a threshold of about $10^{-1}$, after which the force error shoots up. Thus, $10^{-1}$ is the threshold we used in our inference experiments.

**Speedup in simulation**   To test the speedup of DEQuiformer in realistic simulation, we run relaxations based on configurations from OC20. Each sample includes a slab model for the surface and an adsorbate on it as an initial guess. Starting from 100 samples of the OC20 200k train set we run 100 relaxation steps each to get the lowest energy geometry. We use the same checkpoint as in figure 3b and table 2b and compare a one-layer DEQuiformer to a 14-layer EquiformerV2, since the latter is the closest to DEQuiformer's accuracy. The results are summarized in 2a. DEQuiformer is faster than EquiformerV2 in practical scenarios, while being much smaller and more accurate on the

(a)

| OC20 Relaxation | FP reuse | $\epsilon_{test}^{FPreuse}$ | Time [s] | # Solver steps |
|---|---|---|---|---|
| EquiformerV2 (14 layers) | | | $12.92 \pm 0.26$ | - |
| DEQ (1 layer) | ✗ | ✗ | $32.98 \pm 0.41$ | $29.37 \pm 7.03$ |
| DEQ (1 layer) | ✓ | ✗ | $20.37 \pm 0.43$ | $18.05 \pm 2.19$ |
| DEQ (1 layer) | ✓ | ✓ | $\mathbf{12.38 \pm 0.33}$ | $\mathbf{11.03 \pm 2.91}$ |

(b)

| OC20 200k | Force | Energy | # Weights |
|---|---|---|---|
| EquiformerV2 (1 layer) | 0.052 | 0.58 | 4.8M |
| EquiformerV2 (4 layers) | 0.045 | 0.52 | 11.7M |
| EquiformerV2 (8 layers) | 0.045 | 0.49 | 21.1M |
| EquiformerV2 (14 layers) | 0.044 | 0.50 | 35.1M |
| DEQ (1 layer) | **0.043** | **0.48** | 4.8M |
| DEQ (2 layers) | **0.042** | **0.47** | 7.1M |

(a) **Speed in relaxation simulation.** DEQuiformer is faster than EquiformerV2 in simulation when fixed-point reuse and a relaxed solver threshold $\epsilon_{test}^{FPreuse}$ (section 3) are combined.

(b) **Accuracy on OC20.** DEQuiformer is more accurate than EquiformerV2 on forces and energy. Force MAE is in units of kcal/mol/Å, energy MAE in kcal/mol. Lower is better.

Table 2: DEQuiformer is (a) faster in relaxation simulations (b) more accurate on OC20.

test set. Both reusing previous fixed-points and relaxing the solver threshold are necessary to gain a speedup, reducing the number of layer evaluations from roughly 29 to 11.

## 5 LIMITATIONS

The training time for DEQuiformer can be about twice as long as for EquiformerV2. This has been an issue for DEQs in general and is caused by the second fixed-point problem in the implicit function theorem. Since DEQuiformer require less memory, longer training times can be offset to a certain extent by increasing the batch size,. We did not increase the batch size in our experiments, though, as it could have obscured the direct comparison to EquiformerV2.

DEQs are fastest if the data has a temporal structure like that of molecular dynamics simulations, such that fixed-points can be reused between time steps. This is usually the case in simulations, since modeling faster changing events requires proportionally smaller time steps. If only individual configurations are evaluated, DEQs are slower, but still might offer higher accuracy and smaller model sizes, as on the OC20 benchmark.

While converting EquiformerV2 into a DEQ worked remarkably well, it remains to be tested if every machine learning force field can be "DEQuified". Specifically, EquiformerV2 predicts forces via a separate output head, while some models compute forces as gradients of the energy. EquiformerV2 and others demonstrated that predicting the forces directly reaches state-of-the-art accuracy. In some cases, predicting forces through gradients might be preferable since it ensures energy conservation, which can improve MD simulation stability. Predicting forces via gradients in DEQ-style models could simply be done by backpropagating through the solver; however, this would sacrifice the training memory savings that we obtain in the current work. To save memory, one could use the implicit function theorem and potentially similarly warm-starting the gradient fixed-point solver from the previous time step. We leave this to future work.

## 6 CONCLUSION

In this work, we explored the integration of Deep Equilibrium models and machine learning force fields to enhance the efficiency of molecular dynamics (MD) simulations. We "DEQuify" the state-of-the-art model EquiformerV2 with minimal changes by replacing its deep stack of layers with a more compact fixed-point layer. This approach allows us to leverage the temporal similarity between successive MD simulation states by reusing fixed-points, as well as the ability to trade off accuracy and speed. On the MD17 and MD22 datasets, our DEQuiformer model achieves substantial improvements in parameter efficiency and inference speed at similar or better accuracy compared to the original EquiformerV2. On the much larger OC20 200k dataset, DEQuiformer reaches significantly higher accuracy compared to the base model. This suggests a promising new research direction for machine learning force fields, focusing on exploiting the temporal nature of MD simulations to enhance computational efficiency. Since DEQs are in principal orthogonal to the base model, we expect that any improvements in the base architectures or DEQs in the future should complement each other and propel the performance of DEQ force fields even further.

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

# A  APPENDIX

## A.1  ADDITIONAL BACKGROUND

### A.1.1  EQUIFORMERV2 ARCHITECTURE

We provide some further detail on the EquiformerV2 architecture and its three main components: the embedding, the transformer layers, and the output heads. A complete description with additional details on layer norm, multi-head-attention and non-linearities can be found in the original papers Liao & Smidt (2023); Liao et al. (2024).

**Embedding**   An input sample consists of the positions and types of all the atoms in the molecule. The embedding block maps each atom $i$ to a higher dimensional node embedding $\mathbf{h}^{()}i$, consisting of atom and edge-degree embeddings. The edge-degree embeddings transform a constant one vector with an message passing SO(2) layer, multiplied with edge distance embeddings, and aggregated by summing. Edge distance embeddings are the relative distances between the nodes, encoded by a learnable radial function on top of a Gaussian radial basis. This sum is rescaled by a scalar $\alpha$ and added to an linear embedding of the one-hot atom number $\mathbf{z}$:

$$u_t = \alpha \sum_{s \in \mathcal{N}(i)} v(1, 1, r_{is}) \tag{12}$$

$$\mathbf{h}^{()}i = Embed(\mathcal{G})_i = \text{linear}(\text{one-hot}(\mathbf{z}_i)) + u_i \tag{13}$$

$\mathcal{N}(t)$ means the neighbourhood of atom $i$, defined by the set of atoms that are within a user-specified cutoff radius from the atom $i$.

**EquiformerBlock**   We write *EquiformerBlock*$(\mathcal{G})$ to refer to a stack of $L$ Equiformer layers. Each layer consists of equivariant graph attention, layer norm and feed-forward networks. The equivariant graph attention updates the node features $h$ using equivariant messages (equation 3). However, instead of just summing up the messages directly to update a target node, Equiformer weights each message with an attention weight to get the final message which is then summed over all source nodes:

$$m_{ts} = a_{ts} \cdot v_{ts} \tag{14}$$

$$h'_t = h_t + \text{linear}\left(\sum_{s \in \mathcal{N}(t)} m_{ts}\right) \tag{15}$$

The attention weights are calculated using MLP attention Liao & Smidt (2023); Brody et al. (2021) operating only on the rotation invariant $L = 0$ features:

$$z_{ts} = k^\top \text{LeakyReLU}(f(h_t^0, h_s^0)) \tag{16}$$

$$a_{ts} = \frac{\exp(z_{ts})}{\sum_{k \in \mathcal{N}(t)} \exp(z_{tk})} \tag{17}$$

with a learnable weight vector $k$.

**Output Heads**   The output heads take all the node features and process them depending on the type of target. For the energy, the $l = 0$ features of each node are transformed by an MLP and summed together for the final prediction. For the forces, an additional layer of equivariant graph attention is used, and the $l = 1$ features of each atom are directly treated as the prediction for the force.

### A.1.2  INEXACT GRADIENTS IN DEQ

The computational bottleneck in equation 9 is to compute the inverse. Previous work has therefore explored approximating it via its Neumann series, sometimes called the phantom gradient Fung et al. (2022); Geng et al. (2021). Often, keeping only the first term (the identity) is good enough, which leads to the so-called 1-step gradient

$$\frac{\partial L}{\partial \theta} \approx \frac{\partial L}{\partial \mathbf{h}^*} \frac{\partial f_\theta(\mathbf{h}^*, x)}{\partial \theta} \tag{18}$$

The 1-step gradient can be implemented by simply passing the fixed-point through the implicit layer one additional time, this time with tracked gradients using autograd. Many recent works have used the 1-step gradient with great success Cao et al. (2024); Bai et al. (2022); Geng et al. (2023). We found however that while the 1-step gradient leads to 2-3x faster training compared to solving the fixed-point system in equation 10, it resulted in a significant reduction in accuracy, which is why we do not use the 1-step gradient in this paper.

## A.2 METHOD

**Aggregated metric over MD17/MD22**    (paragraph moved from the main text to save space)

We use minmax normalization to rescale the errors of the different models on each molecule to $[0, 1]$, where the models are DEQuiformer and EquiformerV2 with various number of layers $M \in \{DEQ1, DEQ2, E1, E4, E8\}$ . To get summary statistics per model, we then take the mean (Avg) and standard error of the mean (Sem) over all normalized molecules.

$$\text{NormMAE}_M^{mol} = \frac{\text{MAE}_M^{mol} - min_M \left( \text{MAE}_M^{mol} \right)}{max_M \left( \text{MAE}_M^{mol} \right) - min_M \left( \text{MAE}_M^{mol} \right)} \tag{19}$$

$$\text{Avg}_M = \frac{1}{N_{mol}} \sum_{mol} \text{NormMAE}_M^{mol} \tag{20}$$

$$\text{Sem}_M = \frac{1}{\sqrt{N_{mol}}} \sqrt{\frac{1}{N_{mol}} \sum_{mol} \left( \text{NormMAE}_M^{mol} - \text{Avg}_M \right)^2} \tag{21}$$

**Hyperparameters for MD17/MD22**    To facilitate a fair and straightforward comparison, we follow the hyperparameters set out by EquiformerV1 Liao & Smidt (2023) and EquiformerV2 Liao et al. (2024).

Since EquiformerV2 did not evaluate on MD17/MD22, we refer to the EquiformerV1 Liao & Smidt (2023) codebase for training settings, which also provided the training loop for MD17/MD22 of our implementation. To facilitate training on the smaller MD dataset and be economical with our GPU resources, we made our EquiformerV2 significantly smaller than the original settings in Liao et al. (2024). The biggest impact in terms of training and inference speed was due to the smaller maximum feature degree of $l = 3$ (from previously $l = 6$), which was also used in EquiformerV1. We observed that benefits from higher $l$ are neglectable on small datasets like MD17, as Liao et al. (2024) also noted for the similarly sized QM9 dataset. We kept all parameters of the optimizer identical to EquiformerV1.

**Hyperparameters for OC20 S2EF 200k**    EquiformerV2 provides hyperparameters for OC20 2M, which we take as a proxy for the OC20 200k split we train on. The only changes made to the EquiformerV2 model are (1) a reduction in the number of layers down from 12 and (2) limiting the maximum spherical harmonics degree to $l = 3$, since the 200k split is ten times smaller than the 2M split, and because it significantly increases the computational cost. EquiformerV2 made minor changes to the optimizer parameters compared to V1. A full breakdown of hyperparameters is in table 3.

**Implementation**    We use the model of EquiformerV2 from commit fa32143, which depends on open catalyst commit 5a7738f. The open catalyst repository (now called FairChem) has since undergone significant changes. For MD17/MD22 we modify the training loop from the code of EquiformerV1 Liao & Smidt (2023) from commit b7e7a0d. The DEQ solver is adapted from the TorchDEQ library Geng & Kolter (2023).

## A.3 ADDITIONAL RESULTS

**Energy on MD17/MD22**    For molecular dynamics the focus is on accurate forces rather then energies. This is also reflected in the much higher weighting of the loss terms of the forces $\lambda_f = 80$ to $\lambda_e = 1$ that was used in EquiformerV2, and common in the field Liao et al. (2024); Christensen & von Lilienfeld (2020). For completeness we include the energies in in table 4.

| Hyperparameters | MD17/MD22 | OC20 S2EF 200k |
|---|---|---|
| Optimizer | AdamW | |
| Learning rate scheduling | Cosine with linear warmup | |
| Warmup epochs | 10 | 0.1 |
| Initial learning rate | $1 \times 10^{-6}$ | $4 \times 10^{-5}$ |
| Maximum learning rate | $5 \times 10^{-4}$ | $2 \times 10^{-4}$ |
| Minimum learning rate | $1 \times 10^{-6}$ | $2 \times 10^{-6}$ |
| Number of epochs | 500 | 3 |
| Batch size | 4 | |
| Force loss metric | L2 MAE | L2 MAE |
| Energy loss metric | L2 MAE | L1 MAE |
| Force loss weight $\lambda_F$ | 80 | 100 |
| Energy loss weight $\lambda_E$ | 1 | 2 |
| Number of layers | 1, 4, 8 EquiformerV2 / 1, 2 DEQ | |
| Weight decay | $5 \times 10^{-3}$ | $10^{-3}$ |
| Gradient norm clipping | 1000 | 100 |
| Dropout rate (alpha dropout) | 0.1 EquiformerV2 / 0 DEQ | |
| Stochastic depth (path dropout) | 0.05 | |
| Cutoff radius (Å) | 5.0 | 12 |
| Maximum number of neighbors | 500 | 20 |
| Number of radial bases | 128 | |
| Maximum degree $l_{max}$ | 3 | |
| Maximum order $M_{max}$ | 2 | |
| Grid resolution of point samples $R$ | 14 | |
| Hidden dimension in feed forward networks $d_{ffn}$ | 128 | |
| Dimension of hidden scalar features in radial functions $d_{edge}$ | 32 | 128 |
| Embedding dimension (spherical channels) $d_{embed}$ | 64 | 128 |
| $f_{ij}^{(L)}$ dimension $d_{attn\_hidden}$ | 16 | 64 |
| Number of attention heads $h$ | 4 | 8 |
| $f_{ij}^{(0)}$ dimension $d_{attn\_alpha}$ | 16 | 64 |
| Value dimension $d_{attn\_value}$ | 4 | 16 |
| DEQ root solver | Anderson | |
| Maximum number of forward steps (stopping criterion) | 40 | |
| Absolute error tolerance (stopping criterion) training $\epsilon_{train}$ | $10^{-3}$ | |
| Absolute error tolerance (stopping criterion) inference $\epsilon_{test}$ | $10^{-1}$ | |
| Fixed-point correction loss terms | 3 | |

Table 3: Hyperparameters for EquiformerV2 and DEQuiformer. Training hyperparameters for MD17/MD22 are taken from the EquiformerV1 codebase. Model parameters are reduced to roughly a quarter to match the smaller MD17/MD22 benchmark. For OC20 training and model settings are taken from the EquiformerV2 repository.

| MD17 | Ethanol | Malonaldehyde | Benzene | Uracil | Toluene | Salicylic acid | Naphthalene | Aspirin |
|---|---|---|---|---|---|---|---|---|
| | Energy | Energy | Energy | Energy | Energy | Energy | Energy | Energy |
| EquiformerV2 (1 layer) | 0.899 | 0.567 | 0.253 | 0.844 | 1.352 | 1.290 | 0.536 | 1.704 |
| EquiformerV2 (4 layers) | **0.25** | **0.34** | **0.17** | 0.42 | **0.29** | **0.60** | **0.27** | **0.87** |
| EquiformerV2 (8 layers) | **0.26** | **0.33** | 0.22 | **0.34** | **0.25** | 0.54 | **0.25** | 0.80 |
| DEQ (1 layer) | 0.31 | 0.38 | 0.17 | **0.42** | 0.35 | 0.79 | 1.27 | 1.13 |
| DEQ (2 layers) | 0.27 | 0.36 | **0.17** | 0.36 | 0.44 | **0.49** | 0.29 | **0.76** |

| MD22 | Ac-Ala3-NHMe | DHA | AT-AT | Stachyose | AT-AT-CG-CG | Buckyball catcher | # Weights |
|---|---|---|---|---|---|---|---|
| | Energy | Energy | Energy | Energy | Energy | Energy | |
| EquiformerV2 (1 layer) | 2.585 | 3.306 | 5.936 | 4.168 | 6.113 | 3.670 | 670k |
| EquiformerV2 (4 layers) | 1.85 | 2.96 | 6.46 | 3.19 | 6.09 | **3.28** | 1.7M |
| EquiformerV2 (8 layers) | **1.38** | 3.54 | 3.55 | **2.52** | **5.76** | 3.85 | 3M |
| DEQ (1 layer) | **1.41** | **1.94** | **4.79** | **2.53** | 5.99 | 4.04 | **670k** |
| DEQ (2 layers) | 1.41 | **1.49** | **3.52** | 2.53 | 5.81 | **2.55** | **1M** |

Table 4: **Accuracy of energy prediction on MD17/MD22.** Numbers denote mean average error (MAE) on the test set. Lower is better. Energies are in units of kcal/mol.

**Training run on OC20** In figure 2 of the main text we depicted that DEQuiformer achieves lower train and test error than EquiformerV2 throughout training on Aspirin. For completion we also plot the training run for OC20 200k in figure 5. Note that the choppy behaviour of the training curve is due to resets of averaging statistics after each epoch.

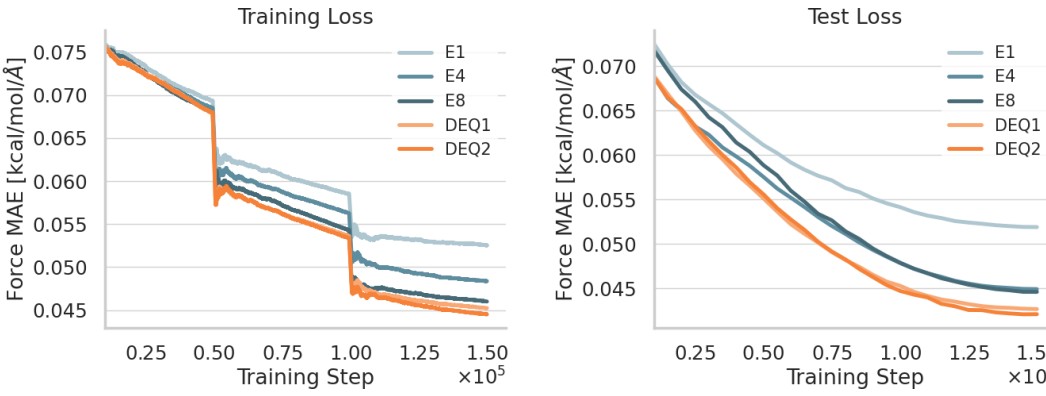

(a) DEQuiformer trains faster, achieving lower train error. We plot the error averaged over the current epoch. The step-like jumps are due to resetting the average at the start of a new epoch.

(b) Lower train error translates to lower test error.

Figure 5: DEQuiformer enjoys stable training dynamics, reaching lower train and test error than EquiformerV2 on OC20 200k.

**Fixed-point reuse approximately preserves Markovianity**   An important property of molecular dynamics is that the forces only depend on the current state, known as the Markovian property. To test if reusing fixed-points breaks Markov property, we compare the predicted forces $\mathbf{F}$ with and without fixed-point reuse. At each timestep we calculate the relative difference in the forces as

$$\Delta F_{rel}^{atom}(atom\ i) = \frac{|\mathbf{F}_i^{fpr} - \mathbf{F}_i|_x}{\frac{1}{2}\left(|\mathbf{F}_i^{fpr}|_x + |\mathbf{F}_i|_x\right)} \tag{22}$$

$$\Delta F_{rel}^{sample}(sample\ j) = \frac{1}{N} \sum_{i\in atoms}^{N} \left(\Delta F_{rel}^{atom}(i)\right) \tag{23}$$

$$\Delta F_{rel} = \frac{1}{M} \sum_{j\in test}^{M} \left(\Delta F_{rel}^{sample}(j)\right) \tag{24}$$

where $|\cdot|_x$ denotes the l2-norm over the three spatial components of a force vector on one atom $i$. We run and average over $M = 1k$ consecutive samples of Aspirin from the MD17 dataset. The relative force difference $\Delta F_{rel}$ is depicted in figure 6. We see a deviation in the predicted forces between starting from zero initialization and from the previous fixed-point of, on average, $0.4\%$. The deviation remains constant over time. We repeat the experiment for the 100 times 100 relaxation steps reported in section 4.1, and measured a deviation of $0.8\%$. The deviation is much smaller than the average prediction error, so we conclude that fixed-point reuse approximately preserves the Markov property.

**Scaling compute**   The paper directly compares DEQuiformer against EquiformerV2. We do so with limited compute compared to the EquiformerV2 paper Liao et al. (2024), which trained up to 135M parameters on a larger datapslit (200k vs >100M) for >1500 GPU days.

To demonstrate that our results are robust, we scale up selected runs. In figure 7a we train Aspirin with double the epochs as in the paper (1k as opposed to 500) at increasing model sizes. The smallest datapoint (left) is the same model size that we used in the main text for MD17/MD22, and the largest (right) the same as previously used for OC20. Note that at the same width DEQuiformer has much fewer total parameters, e.g. DEQ1∼4.8M compared to E8∼21M for the right-most width. The accuracy gap between DEQuiformer to EquiformerV2 remains when scaling the model size.

We report the scaling with an increase in training epochs on OC20 200k in figure 7b. We did not scale up the model size, as EquiformerV2 would run out of memory. Instead we depict the same

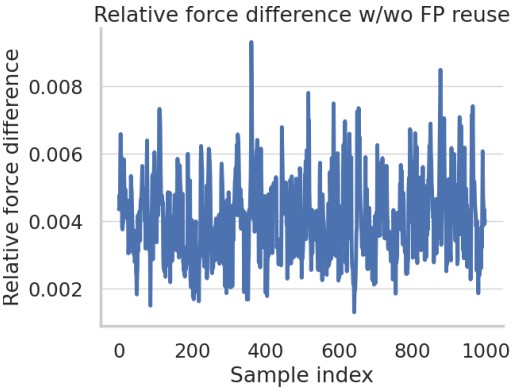

Figure 6: **Markov property.** Initializing from the previous fixed-point, compared to initializing from zero, leads to very small deviation in forces $\Delta F_{rel}$ below one percent. This means, initialization from the past fixed point has almost no effect on the accuracy of the prediction.

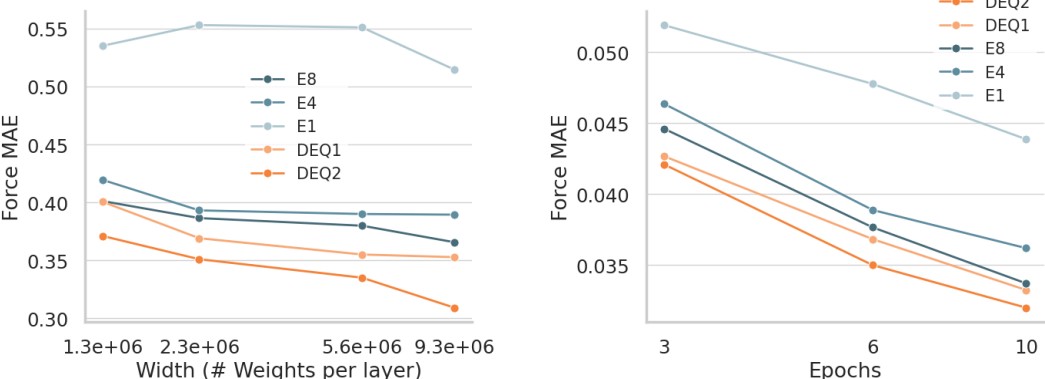

(a) Error scaling with model width on Aspirin (MD17), trained at 1k epochs (compared to 500 in the main text). On the x-axis is the number of parameters per layer. This means that the 8-layer Equiformer has 8 times the parameter compared to the 1-layer DEQuiformer.

(b) Error scaling with more epochs on OC20 200k. All models are getting better with more epochs, but DEQuiformer remains the leader in accuracy over Equiformer.

Figure 7: Error scaling with more epochs and model size.

model size as in paper for OC20, with the left-most data point also trained on the same number of epochs. Again DEQuiformer increase in accuracy is robust when scaling up.

Compared to Liao et al. (2024) there is still a large gap in compute used, so more work is needed to verify that the results hold at truly large scales outside of our budget. Nevertheless our initial experiments seem robust up to the scales tested.

**Pseudocode** To clarify our algorithm, we provide pseudocode for DEQuiformer in algorithm 2 as well as for the original DEQ Bai et al. (2019) in algorithm 1.

The original DEQ paper (algorithm 1) is based on a transformer acting on a sequence of language tokens. $x_{1:T}$ denote the input sequence and $y_{1:T}$ the output sequence of tokens. $f_\theta$ is a (weight tied) transformer layer.

DEQuiformer (algorithm 2) acts on a cloud of atom positions and atom types. We drop the token indices $\cdot_{1:T}$ and omit the atom indices for readability. The `BackwardDEQ` procedure remains the same. The the predicted and ground truth labels $y$ each consist of forces and the energy instead of sequences. We made a couple of changes to the original DEQ. The original DEQ paper Bai et al. (2019) used a linear initialization of the input injection, whereas we use EquiformerV2's encoder. We

also added a decoder (EquiformerV2's force and energy prediction heads). The solver is similar, but we use Anderson acceleration instead of Broyden's method, where $\beta$ is the mixing parameter, $c_j$ are coefficients determined by minimizing the residuals, and $m$ is the number of previous iterations used in the mix Anderson (1965); Geng & Kolter (2023). We also add a normalization after each input injection. The original DEQ initialized fixed-points as zeros, whereas we took inspiration from Bai et al. (2022) and initialized with the previous fixed-point during inference. From [Bai et al. (2022) we also take the fixed-point correction loss and the relaxed solver tolerance $\epsilon$. The main change we made to EquiformerV2 was to remove alpha dropout as it hurt performance and replace path dropout with a recurrent path dropout (not shown in the algorithm).

---

**Algorithm 1** Deep Equilibrium Model (DEQ), Bai 2019

---

1: **procedure** DEQ($\hat{x}_{1:T}, \theta, \epsilon$)
2:   Define $g_\theta(z_{1:T}; \hat{x}_{1:T}) = f_\theta(z_{1:T} + \hat{x}_{1:T}) - z_{1:T}$
3:   Initialize $z_{1:T}^{(0)} \leftarrow 0$
4:   $i \leftarrow 0$
5:   **while** $\|g_\theta(z_{1:T}^{(i)}; \hat{x}_{1:T})\| > \epsilon$ **do**                      ▷ fixed-point solver
6:     $z_{1:T}^{(i+1)} \leftarrow z_{1:T}^{(i)} - \alpha B g_\theta(z_{1:T}^{(i)}; \hat{x}_{1:T})$           ▷ Broyden's method
7:     $i \leftarrow i + 1$
8:   **end while**
9:   $z_{1:T}^* \leftarrow z_{1:T}^{(i)}$
10:   **return** $z_{1:T}^*$
11: **end procedure**
12:
13: **procedure** BACKWARDDEQ($z^*, y_{pred}, y_{gt}, \theta, \epsilon$)
14:   Compute $\frac{\partial \mathcal{L}}{\partial z^*}$ using the loss function $\mathcal{L}(y_{pred}, y_{gt})$
15:   Solve the linear system (IFT, second fixed-point solver):

$$\left(J_{g_\theta}^\top \big|_{z^*}\right) x + \left(\frac{\partial \mathcal{L}}{\partial z^*}\right)^\top = 0$$

16:   Compute the gradient:

$$\frac{\partial \mathcal{L}}{\partial \theta} = -\left(\frac{\partial \mathcal{L}}{\partial z^*}\right)\left(J_{g_\theta}^{-1}\big|_{z^*}\right)\frac{\partial f_\theta}{\partial \theta}$$

17:   **return** $\frac{\partial \mathcal{L}}{\partial \theta}$
18: **end procedure**
19:
20: **procedure** USEDEQ($x_{1:T}, y_{1:T}, \theta, \epsilon, \alpha$)
21:   **while** not done **do**
22:     $\hat{x}_{1:T} \leftarrow W^T x_{1:T}$                      ▷ input injection
23:     $z_{1:T}^* \leftarrow$ DEQ($\hat{x}_{1:T}, \theta, \epsilon$)
24:     $y_{pred} \leftarrow z_{1:T}^*$                      ▷ no decoder
25:     **if** inference **then**
26:       $\frac{\partial \mathcal{L}}{\partial \theta} \leftarrow$ BackwardDEQ($z_{1:T}^*, y_{pred}, y_{1:T}, \theta, \epsilon$)
27:       Update $\theta \leftarrow$ optimizer($\theta, \frac{\partial \mathcal{L}}{\partial \theta}$)
28:     **end if**
29:   **end while**
30:   **return** $\theta$
31: **end procedure**

---

---

**Algorithm 2** DEQuiformer

---

1: **procedure** DEQ$(\hat{x}, \theta, \epsilon, z_{t-1}^*)$
2:     Define $g_\theta(z; \hat{x}) = f_\theta\left((z + \hat{x})\frac{||\hat{x}||}{||z+\hat{x}||} - z\right)$              ▷ added normalization
3:     Initialize $z^{(0)} \leftarrow 0$                           ▷ if training
4:     **if** inference **then**
5:         Initialize $z^{(0)} \leftarrow z_{t-1}^*$                ▷ fixed-point reuse
6:     **end if**
7:     $i \leftarrow 0$
8:     $\{z^{(i)}\} \leftarrow \{\}$                  ▷ intermediate fixed-points for correction loss
9:     **while** $\|g_\theta(z^{(i)}; \hat{x})\| > \epsilon$ **do**
10:         $z^{(i+1)} \leftarrow (1 - \beta)g(z^{(i)}; \hat{x}) + \beta \sum_{j=0}^m c_j z^{(i-j)}$     ▷ Anderson acceleration
11:         **if** training **then**
12:             $\{z^{(i)}\}$ append $z^{(i+1)}$        ▷ if $i$ in $\mathcal{I}$, save intermediate fixed-point
13:         **end if**
14:         $i \leftarrow i + 1$
15:     **end while**
16:     $z^* \leftarrow z^{(i)}$
17:     **return** $z^*, \{z^{(i)}\}$
18: **end procedure**
19:
20: **procedure** USEDEQ$(x, (\mathbf{F}_{gt}, E_{gt}), \theta, \epsilon, \alpha)$
21:     $z_{t-1}^* \leftarrow 0$                  ▷ if inference, save previous fixed-point
22:     **while** not done **do**
23:         $\hat{x} \leftarrow \text{Enc}(x)$            ▷ input injection via Equiformer encoder
24:         $z^*, \{z^{(i)}\} \leftarrow \text{DEQ}(\hat{x}, \theta, \epsilon, z_{t-1}^*)$
25:         $z_{t-1}^* \leftarrow z^*$                ▷ save for fixed-point reuse
26:         $\mathbf{F} \leftarrow \text{Dec}_F(z)$              ▷ Eqiformer decoder
27:         $E \leftarrow \text{Dec}_E(z)$               ▷ Eqiformer decoder
28:         **if** training **then**
29:             $\frac{\partial \mathcal{L}}{\partial \theta} \leftarrow \text{BackwardDEQ}(z^*, (\mathbf{F}, E), (\mathbf{F}_{gt}, E_{gt}), \theta, \epsilon)$
30:             **for** $z^{(i)}$ in $\{z^{(i)}\}$ **do**          ▷ sparse fixed-point correction loss
31:                 $\frac{\partial \mathcal{L}}{\partial \theta} += \text{BackwardDEQ}(z^{(i)}, (\mathbf{F}, E), (\mathbf{F}_{gt}, E_{gt}), \theta, \epsilon)$
32:             **end for**
33:             Update $\theta \leftarrow \text{optimizer}(\theta, \frac{\partial \mathcal{L}}{\partial \theta})$
34:         **end if**
35:     **end while**
36:     **return** $\theta$
37: **end procedure**

---

