# OpenReview forum: "DEQuify your force field: Towards efficient simulations using deep equilibrium models"
_ICLR.cc/2025/Conference — Submitted to ICLR 2025_

### Official Review · Reviewer_edd9 · 2024-10-28

**Soundness:** 2
**Presentation:** 2
**Contribution:** 2
**Rating:** 3
**Confidence:** 5

**Summary:**

This work utilizes deep equilibrium models to enhance the accuracy, speed, and memory efficiency of EquiformerV2. Furthermore, DEQuiformer could accelerate molecular dynamics simulations by leveraging the similarity between successive states.

**Strengths:**

* This paper proposes incorporating DEQ into MLFF.
* Compared to EquiformerV2, DEQuiformer shows advantages.
* The paper demonstrates experiments on the rapid convergence of DEQuiformer's fixed points.

**Weaknesses:**

The related issues will be elaborated in the "questions" section.

**Questions:**

* Lack of Innovation: Although this paper is, to my knowledge, the first to combine DEQ with MLFF, the concept of DEQ is based on the fixed-point property of neural network hidden states. This work seems to merely change the application scenario to MLFF without actual innovation.
* In the introduction, the paper misses a lot of related works in the field, such as [1]-[6].
* The paper highlights that in molecular dynamics simulations, because consecutive frames have similar configurations, the fixed point from the previous frame can be used as the initial trial for the next frame, thus accelerating the process. My concerns are as follows:
1. The experiments provided to support this point are insufficient. For example, it would be useful to compare the number of iterations with and without reuse. Additionally, molecular dynamic simulations should be performed to claim your points.
2. The paper mentions using a validation set of 50 samples from MD17, but Fig. 4b states it uses the validation set. Were these 50 samples selected in consecutive order? Additionally, the paper mentions selecting an extra 1000 consecutive samples. If these were used, they should not be considered part of the validation set. The authors should clarify to avoid reader confusion.
3. Fig. 4b is confusing. What is the percentage relative to? The figure needs a clearer explanation.
4. Is the acceleration effect of DEQuiformer primarily due to reuse or the reduced number of model layers? I suspect the latter is the main factor.
* In conventional MLFF experiments, the MD17 test set includes all remaining data, while rmd17 provides a specific test set. However, this paper uses only 1000 data points from the MD17 test set, which is not a fair comparison.
* The paper claims that the data split for MD22 is the same as for MD17, which is inconsistent with the original MD22 paper [7] and conventional MLFF splits[5,6,8,9].

* The paper lacks results from other benchmark models for MD17 and MD22. Benchmarking with additional methods could provide a better evaluation of the results.

* Why aren’t the energy and force tables for MD17 and MD22 combined? This makes it inconvenient for readers to compare the results. Additionally, the energy results for EquiformerV2 are exceedingly poor. Was the training fully converged? This could imply a bias in lowering the benchmark, making the comparison unfair.

* I greatly appreciate that the authors included a limitations section, but I have some questions regarding certain points mentioned. The limitations section mentions uncertainty about whether DEQ is applicable to other MLFFs because EquiformerV2 uses a separate force output head. I don’t quite understand the logic here. Is DEQuiformer based on the assumption that forces have a fixed-point property? It seems to me that DEQuiformer assumes the hidden layers of molecular representations have a fixed-point property, so DEQ should theoretically be applicable to any model predicting any property, even if forces are obtained via autograd.

* The presentation of the paper could be improved. For readers not very familiar with the field, more introductory content on concepts like equivariance and Equiformer should be added in the preliminary section.

[1] Schütt, Kristof T., et al. "Schnet–a deep learning architecture for molecules and materials." The Journal of Chemical Physics 148.24 (2018).

[2] Gasteiger, Johannes, Janek Groß, and Stephan Günnemann. "Directional Message Passing for Molecular Graphs." International Conference on Learning Representations.

[3] Schütt, Kristof, Oliver Unke, and Michael Gastegger. "Equivariant message passing for the prediction of tensorial properties and molecular spectra." International Conference on Machine Learning. PMLR, 2021.

[4] Coors, Benjamin, Alexandru Paul Condurache, and Andreas Geiger. "Spherenet: Learning spherical representations for detection and classification in omnidirectional images." Proceedings of the European conference on computer vision (ECCV). 2018.

[5] Wang, Yusong, et al. "Enhancing geometric representations for molecules with equivariant vector-scalar interactive message passing." Nature Communications 15.1 (2024): 313.

[6] Wang, Zun, et al. "Efficiently incorporating quintuple interactions into geometric deep learning force fields." Advances in Neural Information Processing Systems 36 (2024).

[7] Chmiela, Stefan, et al. "Accurate global machine learning force fields for molecules with hundreds of atoms." Science Advances 9.2 (2023): eadf0873.

[8] Kovács, Dávid Péter, et al. "Evaluation of the MACE force field architecture: From medicinal chemistry to materials science." The Journal of Chemical Physics 159.4 (2023).

[9] Li, Yunyang, et al. "Long-Short-Range Message-Passing: A Physics-Informed Framework to Capture Non-Local Interaction for Scalable Molecular Dynamics Simulation." The Twelfth International Conference on Learning Representations.

---

> ### Author Response · Authors · 2024-11-19
>
> > Lack of Innovation: Although this paper is, to my knowledge, the first to combine DEQ with MLFF, the concept of DEQ is based on the fixed-point property of neural network hidden states. This work seems to merely change the application scenario to MLFF without actual innovation.
>
> As mentioned in the review, the paper is the first to apply DEQ to MLFF.
> The main insight of the paper is to use the sequential nature of molecular dynamics simulation to achieve faster inference, better accuracy, and smaller models, which we think are valuable contributions to the field.
> We also added further experiments to demonstrate these benefits.
>
>
> &nbsp;
>
> > In the introduction, the paper misses a lot of related works in the field, such as [1]-[6].
>
> We thank the reviewer for pointing these out and have added them to the paper.
>
> [4] references image classification; does the reviewer refer to the similarly-called SphereNet in Liu, Yi, et al. "Spherical message passing for 3d molecular graphs." International Conference on Learning Representations (ICLR). 2022”?
>
> &nbsp;
>
> > The experiments provided to support this point are insufficient. For example, it would be useful to compare the number of iterations with and without reuse.
>
> > Fig. 4b is confusing. What is the percentage relative to? The figure needs a clearer explanation.
>
> We clarified the caption. Figure 4b tests answers the second part of the question: We ran inference with and without fixed point reuse on Aspirin over the test set and plotted the percentage of test points that required $n$ number of fixed point solver steps, where $n$ is plotted on the x-axis.
> To reinforce this point, we conducted additional experiments for an MD simulation on OC20; see below for details.
>
>
> &nbsp;
>
> >  Additionally, molecular dynamic simulations should be performed to claim your points.
>
> Thank you for the constructive suggestion. We added MD simulations of relaxations for OC20 data with and without fixed point reuse: We took 100 random points as starting points and ran the relaxation for 100 steps each. The average number of fixed point iterations went from 29 to 11 when incorporating fixed point reuse and the relaxed solver threshold. This makes DEQuiformer faster than a 14-layer EquiformerV2, while DEQuiformer’s accuracy is much better on OC20, as seen in Figure 3 b.
>
>
>
>
> &nbsp;
>
> > Is the acceleration effect of DEQuiformer primarily due to reuse or the reduced number of model layers? I suspect the latter is the main factor.
>
> To clarify the internals of a DEQ: Each iteration of the fixed point solver costs as much as one layer in a standard neural network. A DEQ with 1-layer taking 10 iterations is roughly equivalent to a 10-layer standard neural network.
> Without fixed point reuse, our DEQ would be as expensive as a standard neural network with the same accuracy. However, fixed point reuse cuts the number of solver steps significantly, as is shown in Figure 4b and our additional OC20 MD simulation results, making the DEQ faster while also being more accurate.
>
>
> &nbsp;
>
> > In conventional MLFF experiments, the MD17 test set includes all remaining data, while rmd17 provides a specific test set. However, this paper uses only 1000 data points from the MD17 test set, which is not a fair comparison.
> The paper claims that the data split for MD22 is the same as for MD17, which is inconsistent with the original MD22 paper [7] and conventional MLFF splits[5,6,8,9].
>
> > The paper lacks results from other benchmark models for MD17 and MD22. Benchmarking with additional methods could provide a better evaluation of the results.
>
> We understand the reviewer's concern regarding the evaluation protocol. We adopted it from the codebase of EquiformerV1, which used 1,000 random test points for MD17, and changed it to 1,000 uniformly spaced samples for better coverage. In our test on Aspirin, the deviation between testing on the full dataset of 500,000 points compared to our 1,000 points was 2%, while being much faster. For simplicity, we also adopted the same protocol for MD22. We agree that this makes the numbers less directly comparable with other models. However, we benchmarked both EquiformerV2 and DEQuiformer on the identical test set, ensuring a fair, controlled comparison between these models. As our focus was on the relative performance gains from introducing the DEQ mechanism, we did not aim to provide a comprehensive benchmark against other existing models.

---

> > ### Author Response · Authors · 2024-11-23
> >
> > Update:
> >
> > We reran the evaluation of MD17 and MD22 on the full test set, following the reviewers suggestion. The updated numbers are reflected in the table of the revised manuscript. The numbers barely changed, and the advantage of DEQuiformer remains consistent.

---

> > > ### Comment · Reviewer_edd9 · 2024-11-25
> > >
> > > Thank you very much for the response and the additional experiments. However, I still have the following concerns:
> > >
> > > * **Lack of Innovation:** While I did acknowledge that "the paper is the first to apply DEQ to MLFF," this does not necessarily mean that the work is highly innovative. For instance, if you could demonstrate the existence of such a fixed point in MLFF and solve this issue using DEQ, that would be quite interesting. However, this work merely applies a method from a 2019 paper to MLFF, which, in my opinion, lacks sufficient innovation to be published at a top-tier conference in 2024.
> > > * **Paper Citation:** This was an oversight on my part.
> > > * **Fig. 4b:** Thank you for explaining this figure. I still have a question: the bars of the same color do not seem to sum up to 100%. For example, "fixed point reuse" is quite prominent. Why is this the case?
> > > * **Molecular Dynamic Simulations:** My concern was to illustrate the acceleration brought by DEQ through molecular dynamics simulations, not accuracy. So, I guess you meant Fig. 3a rather than the added Figure 3b. You might think that inference time reflects the acceleration in molecular dynamics simulations, but I would like to clarify this further. Since you claimed the relevance of molecular dynamics simulations in your paper, I suggest you should include experiments related to molecular dynamics simulations.
> > > * **MD17 and MD22:** Firstly, although the authors claim to have "benchmarked both EquiformerV2 and DEQuiformer on the identical test set, ensuring a fair, controlled comparison between these models," adding other models benchmarked on the same dataset in the table would more clearly illustrate the effectiveness, including whether your own trained Equiformer results are reasonable. Secondly, I still suggest presenting energy and force in a single table as this is the standard comparison method in MLFF-related literature, which improves readability. Thirdly, I noticed you updated the energy table. The supervised energy results have improved compared to the previous version, but anyone familiar with the state-of-the-art results for MD17 and MD22 can see that these results are still an order of magnitude larger than the state-of-the-art. Therefore, comparisons on models that are not fully trained are meaningless.
> > > * **Caption:** Thank you for the additional captions for the figures and tables. However, I suggest further improving the quality of the paper. For instance, the captions should clearly explain what is explicitly shown in the figures, as well as the meaning of each symbol and the different colors. In the current version, many figures require me to guess what you are trying to convey rather than understanding it through the captions. Your captions now primarily state the conclusions that can be drawn from the figures, rather than explaining the figures themselves.

---

> > > > ### Author Response · Authors · 2024-11-25
> > > >
> > > > Thank you very much for the thoughtful reply.
> > > >
> > > > &nbsp;
> > > >
> > > > > Lack of Innovation: While I did acknowledge that "the paper is the first to apply DEQ to MLFF," this does not necessarily mean that the work is highly innovative. For instance, if you could demonstrate the existence of such a fixed point in MLFF and solve this issue using DEQ, that would be quite interesting. However, this work merely applies a method from a 2019 paper to MLFF, which, in my opinion, lacks sufficient innovation to be published at a top-tier conference in 2024.
> > > >
> > > > Thank you for expanding on your perspective. We want to emphasise that the work combines techniques from [1], [2], and others in a non-straightforward way. Additional changes like input injection normalization and recurrent dropout were necessary to achieve the reported speedup and accuracy gains. Additionally, to the best of our knowledge, the concept of exploiting temporal continuity in MD simulations is completely new to the field.
> > > >
> > > > [1] [1909.01377] Deep Equilibrium Models
> > > >
> > > > [2] [2204.08442] Deep Equilibrium Optical Flow Estimation
> > > >
> > > > &nbsp;
> > > >
> > > > > Fig. 4b: Thank you for explaining this figure. I still have a question: the bars of the same color do not seem to sum up to 100%. For example, "fixed point reuse" is quite prominent. Why is this the case?
> > > >
> > > > To clarify, we plot two distributions: With fixed-point reuse (orange) and without fixed-point reuse (blue). Both distributions add up to 100% individually. Especially for the blue bars, there is also a long tail of sub-1% contributions that do not appear in the plot because they are individually too small to be visible.
> > > >
> > > > &nbsp;
> > > >
> > > > > Molecular Dynamic Simulations: My concern was to illustrate the acceleration brought by DEQ through molecular dynamics simulations, not accuracy. So, I guess you meant Fig. 3a rather than the added Figure 3b. You might think that inference time reflects the acceleration in molecular dynamics simulations, but I would like to clarify this further. Since you claimed the relevance of molecular dynamics simulations in your paper, I suggest you should include experiments related to molecular dynamics simulations.
> > > >
> > > > We apologise for our confusing response. We ran MD simulations for OC20 systems and provided the timings in Table 2 a). As can be seen, DEQ is faster than Equiformer in actual simulations, if both fixed point reuse and the relaxed solver threshold are used.
> > > > We mentioned Fig. 3b to illustrate that we are faster while also having better accuracy.
> > > >
> > > > &nbsp;
> > > >
> > > > > MD17 and MD22: Firstly, although the authors claim to have "benchmarked both EquiformerV2 and DEQuiformer on the identical test set, ensuring a fair, controlled comparison between these models," adding other models benchmarked on the same dataset in the table would more clearly illustrate the effectiveness, including whether your own trained Equiformer results are reasonable. Secondly, I still suggest presenting energy and force in a single table as this is the standard comparison method in MLFF-related literature, which improves readability. Thirdly, I noticed you updated the energy table. The supervised energy results have improved compared to the previous version, but anyone familiar with the state-of-the-art results for MD17 and MD22 can see that these results are still an order of magnitude larger than the state-of-the-art. Therefore, comparisons on models that are not fully trained are meaningless.
> > > >
> > > > Why the energy errors are so poor is not quite clear since the original EquiformerV2 work never published any experiments on MD17/22. However, the force errors are comparable to other methods, which shows that the model is trained to a meaningful point. Further scaling in model size and training time improves all models as shown in Figure 7a), but the advantage of DEQuiformer over Equiformer remains and is even somewhat increasing.  For the current work, we are mainly interested in MD simulations and therefore mainly in forces and not energies though.
> > > >
> > > > &nbsp;
> > > >
> > > > > Caption: Thank you for the additional captions for the figures and tables. However, I suggest further improving the quality of the paper. For instance, the captions should clearly explain what is explicitly shown in the figures, as well as the meaning of each symbol and the different colors. In the current version, many figures require me to guess what you are trying to convey rather than understanding it through the captions. Your captions now primarily state the conclusions that can be drawn from the figures, rather than explaining the figures themselves.
> > > >
> > > > Thank you for the feedback. We will further work on clarifying the figure captions to be more descriptive for the camera ready version.

---

> ### Author Response · Authors · 2024-11-19
>
> > The paper mentions using a validation set of 50 samples from MD17, but Fig. 4b states it uses the validation set. Were these 50 samples selected in consecutive order? Additionally, the paper mentions selecting an extra 1000 consecutive samples. If these were used, they should not be considered part of the validation set. The authors should clarify to avoid reader confusion.
>
> Thank you for pointing this out; we fixed the mistake in the paragraph referring to figure 4b. We used the 1,000 point test set, not the validation set, to create the figure. We will correct this in our paper. The validation data was also randomly sampled, not consecutive and used only for early stopping (which was always the last checkpoint and can thus be ignored).
>
>
>
>
> &nbsp;
>
> > Why aren’t the energy and force tables for MD17 and MD22 combined? This makes it inconvenient for readers to compare the results. Additionally, the energy results for EquiformerV2 are exceedingly poor.
>
> We separated forces and timing from energy to improve readability, as a single table would be hard to fit. For OC20 our energy results are significantly worse than reported in EquiformerV2, since EquiformerV2 trained a larger model (~10x parameters) on a bigger data split (200k vs >100M) with more compute (>1500 GPU days). Using the default settings of EquiformerV2, the force loss term is 50 times larger than the energy loss term, which causes the energies to converge well only for very long training times beyond our budget.
> As also pointed out by another reviewer, the energy values in the table for MD22 were faulty. Thank you for reporting this, we fixed the numbers.
>
>
> &nbsp;
>
> > Was the training fully converged? This could imply a bias in lowering the benchmark, making the comparison unfair.
>
> To address these concerns we scaled MD17 Aspirin for double the epochs and up to ten times the parameters, and OC20 to three times the epochs. All models improved as expected, the relative improvement of DEQuiformer over Equiformer is unchanged. We will include these experiments in the appendix.
>
>
> &nbsp;
>
> > I greatly appreciate that the authors included a limitations section, but I have some questions regarding certain points mentioned. The limitations section mentions uncertainty about whether DEQ is applicable to other MLFFs because EquiformerV2 uses a separate force output head. I don’t quite understand the logic here. Is DEQuiformer based on the assumption that forces have a fixed-point property? It seems to me that DEQuiformer assumes the hidden layers of molecular representations have a fixed-point property, so DEQ should theoretically be applicable to any model predicting any property, even if forces are obtained via autograd.
>
> We agree one could simply use autograd to train and deploy DEQ-MLFF, but one would lose the memory benefits during training. As you mentioned, we don't assume any specific fixed-point property for forces, so we expect force prediction via energy to work. We will clarify this in the paper.
>
>
> &nbsp;
>
> > The presentation of the paper could be improved. For readers not very familiar with the field, more introductory content on concepts like equivariance and Equiformer should be added in the preliminary section.
>
> Thank you for the feedback. We added more background information to the introduction of the paper.

---

### Official Review · Reviewer_emZx · 2024-11-03

**Soundness:** 3
**Presentation:** 3
**Contribution:** 3
**Rating:** 6
**Confidence:** 4

**Summary:**

This work develops and demonstrates a strategy for converting a deep network to deep equilibrium models for the task of machine learning inter-atomic potentials. The strategy is interesting and novel compared to other strategies in that area, and leads to improvements in both the accuracy and inference speed. By re-using solutions in the fixed point solver in subsequent time steps in a simulation, the inference speed is further improved. The results are demonstrated on a few diverse datasets including MD17, MD22, and OC20.

**Strengths:**

* The strategy is unlike other current approaches to improve ML for molecular simulations using typical message passing systems.
* The strategy works for both molecules and catalysts
* The performance (accuracy and inference speed) can be improved over a current leading model (EquiformerV2), though the EqV2 baselines were not thoroughly tuned for every experiment
* The authors considered possible edge cases and questions about the impact of the fixed point convergence settings on the final performance.

The authors focus on the benefits of DEQ for accelerating MD simulations by re-using fixed point guesses, but I think there are even more important and influential directions that this paper will unlock.
1. The underlying simulations for the training data are themselves ground state calculations that come from a self-consistent solver for the electronic structure, and there is a strong analogy between the methods in this work and the methods in the underlying solver.
2. A subset of the community has been focusing on computing additional equilibrium properties in a system such as charge distribution, and those methods typically work by using standard message passing GNNs to compute an electronegativity and then solving for an equilibrium distribution of charge (e.g. Wai Ko, Behler et al. Nat Com 2021, or Deng, Cedar et al. Nat. Mat. Int. 2023 among many others). I think there is huge potential (pun intended) for this work to lead to large increases in performance in predicting these other ground state problems that require equilibrium across extended systems. Further, there’s probably connections between this work and other methods to accelerate the convergence of the underlying simulations themselves, perhaps by predicting the electron density itself.

**Weaknesses:**

* As the authors point out, computing gradients of the resulting properties w.r.t. the atomic positions is difficult (but not impossible) and by not addressing this only direct-force models can be improved currently.
* The baseline comparisons use model architectures tuned for larger systems. It would be preferable to compare results for the model here on published baseline models for the original (larger) datasets like OC20-2M so that it is clear the results are not simply due to better tuning of the DEQ-architecture compared to the baseline EqV2 architectures
* The authors state that fixed-point re-use cannot be tested for OC20, but it is not clear to me why. Specifically, OC20 has an MD subset that could be used. Further, the authors could simply run a short-timescale MD simulation using the potential (say in ASE) to compare the inference time savings.
* Large variations in Table 1 are a bit suspicious (see question below) and decrease confidence in the conclusions for MD22.

**Questions:**

1. In Table 1 MD22, the force MAEs seem highly variable and perhaps highlight convergence/training stochasticity rather than intrinsic model differences. For example, in MD22/Stachyose, all EqV2 results are ~11, as well as DEQ (2-layer), but DEQ (1-layer is 0.31, 30X smaller). Similarly Ac-Ala3-NHME has a 10X larger force MAE for EqV2(1-layer). Any AT-AT/DHA have ~10-20X smaller force MAEs for DEQ than EqV2. Can the authors confirm the results in this table and/or improve the training consistency? I understand one of the points of DEQ is better training dynamics, but the Stachyose results suggest even with DEQ there’s significant stochasticity similar to the baseline.

2. How stochastic is the fixed point solver inside of DEQ2? Specifically, what is the likelihood that you find different solutions for the same inputs? This is probably especially important for molecular dynamics, as multiple possible solutions for the local potential energy surface could lead to interesting dynamics, or possibly history-dependent artifacts.

3. The authors might want to consider training a model to predict the partial charges or magnetic moments in the mptrj dataset (similar to CHGNet) to greatly improve the applicability of this work. Obviously this is a significant additional experiment, but if these results were competitive there too I think this work would be much more compelling.

4. Experiments for inference-time speedup for simulations in OC20 are also possible, can the authors shed insight on whether the same speedups are seen there too?

---

> ### Author Response · Authors · 2024-11-19
>
> > As the authors point out, computing gradients of the resulting properties w.r.t. the atomic positions is difficult (but not impossible) and by not addressing this only direct-force models can be improved currently.
>
> Energy–gradient forces could still be implemented relatively easily with autograd but one would lose the memory advantage during training.
> We focus on direct force prediction, because EquiformerV2 is SOTA on OC20, which is one of the biggest and most real-world significant datasets available, and other published works use direct force predictions as well [1, 2, 3, 4, 5]. [4, 5] both argued that direct force prediction works as well or better than energy-gradient-forces for large-scale datasets. We hope, though, that this paper inspires follow-up work on energy-gradient based DEQs.
>
> [1] https://www.nature.com/articles/s41524-021-00543-3
>
> [2] https://www.nature.com/articles/s42256-019-0098-0
>
> [3] https://pubs.aip.org/aip/jcp/article-abstract/156/14/144103/2840972/Graph-neural-networks-accelerated-molecular
>
> [4] https://arxiv.org/pdf/2106.08903
>
> [5] https://arxiv.org/pdf/2204.02782
>
>
>
> &nbsp;
>
>
> > The baseline comparisons use model architectures tuned for larger systems. It would be preferable to compare results for the model here on published baseline models for the original (larger) datasets like OC20-2M so that it is clear the results are not simply due to better tuning of the DEQ-architecture compared to the baseline EqV2 architectures
>
> Using the original model size and the 2M dataset takes about 1400 GPU hours, which is, unfortunately, impossible for our lab to train. Instead, we make the comparison as fair as possible by sweeping over the number of layers for EquiformerV2, which is likely the most important hyperparameter for reduced datasets. We did not tune our DEQ but used the same hyperparameters as for EquiformerV2, so it seems unlikely that the hyperparameters favor DEQuiformer.
> Similar arguments apply to the hyperparameters of MD17/22, which we copied from EquiformerV1 since EquiformerV2 did not experiment on MD17/22.
>
> To address scaling concerns we ran additional experiments with double the epochs, up to ten times the model size on MD17 Aspirin, and three times the epochs on OC20. The results have been added to the appendix. All models got better, the performance difference in favor of DEQuiformer remained constant.
>
>
> &nbsp;
>
> > The authors state that fixed-point re-use cannot be tested for OC20, but it is not clear to me why. Specifically, OC20 has an MD subset that could be used. Further, the authors could simply run a short-timescale MD simulation using the potential (say in ASE) to compare the inference time savings.
>
> > Experiments for inference-time speedup for simulations in OC20 are also possible, can the authors shed insight on whether the same speedups are seen there too?
>
> Thank you for your suggestion, we followed it and ran an MD simulation using our force fields to compare the savings of fixed point reuse and time. For this, we took 100 random points from the OC20 train set as starting points and ran the relaxation for 100 steps. The average number of fixed point iterations went from 29 to 11 for with vs without fixed point reuse. This is faster than the 14-layer EquiformerV2, while the accuracy is much higher, as seen in Figure 3 b.
>
>
> &nbsp;
>
> > Large variations in Table 1 are a bit suspicious (see question below) and decrease confidence in the conclusions for MD22.
>
> > In Table 1 MD22, the force MAEs seem highly variable and perhaps highlight convergence/training stochasticity rather than intrinsic model differences. For example, in MD22/Stachyose, all EqV2 results are ~11, as well as DEQ (2-layer), but DEQ (1-layer is 0.31, 30X smaller). Similarly Ac-Ala3-NHME has a 10X larger force MAE for EqV2(1-layer). Any AT-AT/DHA have ~10-20X smaller force MAEs for DEQ than EqV2. Can the authors confirm the results in this table and/or improve the training consistency? I understand one of the points of DEQ is better training dynamics, but the Stachyose results suggest even with DEQ there’s significant stochasticity similar to the baseline.
>
> Thank you for pointing this out! Indeed the numbers for MD22 in the table were faulty. We fixed them in the revised manuscript. The plots are not affected.

---

> ### Author Response · Authors · 2024-11-19
>
> > How stochastic is the fixed point solver inside of DEQ2? Specifically, what is the likelihood that you find different solutions for the same inputs? This is probably especially important for molecular dynamics, as multiple possible solutions for the local potential energy surface could lead to interesting dynamics, or possibly history-dependent artifacts.
>
> To answer the question, we conducted another experiment: On the test set of Aspirin and our OC20 MD simulations, we predicted every point with and without fixed point reuse. We then measure the relative difference between the two predictions and find that they are still Markovian: The average relative difference of the predictions is below 1% for both, Aspirin and OC20.
>
>
>
> &nbsp;
>
> > A subset of the community has been focusing on computing additional equilibrium properties in a system such as charge distribution, and those methods typically work by using standard message passing GNNs to compute an electronegativity and then solving for an equilibrium distribution of charge (e.g. Wai Ko, Behler et al. Nat Com 2021, or Deng, Cedar et al. Nat. Mat. Int. 2023 among many others). I think there is huge potential (pun intended) for this work to lead to large increases in performance in predicting these other ground state problems that require equilibrium across extended systems.
>
> Thank you for the suggestions; this could be promising future directions! We will include the references in the paper.
>
>
> &nbsp;
>
> > The authors might want to consider training a model to predict the partial charges or magnetic moments in the mptrj dataset (similar to CHGNet) to greatly improve the applicability of this work. Obviously this is a significant additional experiment, but if these results were competitive there too I think this work would be much more compelling.
>
> We appreciate the suggestion, training a model on partial charges or magnetic moments in the MPTRJ dataset would be an interesting application of our approach. However, benchmarking on both OC20 and Materials Project datasets is beyond the scope of this paper. This is largely due to the significant computational and engineering resources required to handle these extensive datasets. We hope our work effectively demonstrates the potential of our proposed approach, and keep MPTRJ on the roadmap for future experiments.

---

> ### Comment · Reviewer_emZx · 2024-11-27
>
> Thanks for the detailed response, new experiments, and update of the table with the correct numbers! I changed my review to 6(marginally above the acceptance threshold).

---

### Official Review · Reviewer_yjG1 · 2024-11-04

**Soundness:** 3
**Presentation:** 3
**Contribution:** 3
**Rating:** 6
**Confidence:** 4

**Summary:**

This paper applies Deep Equilibrium Networks (DEQ) [1] to EquiformerV2 [2], leveraging DEQ’s advantages of significantly reduced memory overhead and faster inference. The authors further enhance DEQ’s fixed-point solution by integrating temporal correlation from MD datasets as an initialization strategy, which would otherwise be computationally prohibitive. To validate their approach, they compare the performance of various EquiformerV2 and DEQ variants on the MD22, MD17, and OC20 benchmarks.

References:

[1]https://arxiv.org/abs/1909.01377

[2]https://arxiv.org/abs/2306.12059

**Strengths:**

1. Equivariant networks based on Transformer architectures are highly computationally intensive, requiring high-end GPUs and often taking multiple days to train. Any efforts to reduce this computational burden are valuable.

2. I commend the authors for applying the Deep Equilibrium Model (DEQ) approach to EquiformerV2. Integrating a fixed-point solver into the Transformer architecture—especially by adapting from TorchDEQ[1] and leveraging temporal correlation—is an impressive accomplishment.

References:

[1] https://github.com/locuslab/torchdeq

**Weaknesses:**

1. Temporal correlation in molecular dynamics (MD) datasets has been used effectively to initialize fixed-point solvers. I like this idea, but it relies on an assumption of ordered data, which breaks down during phase transitions. For instance, in the simple case of water melting, the transformation from solid to liquid occurs abruptly, disrupting the expected order and causing this approach to fail at the transition point.


2. This approach also struggles with datasets that lack a clear temporal order, where entries are jumbled and any sense of sequence is lost (as noted by the authors for datasets like rMD17 and OC20). This limitation restricts the applicability of the method, impacting one of the core contributions of this work. Can you please discuss potential ways to address this limitation or expand on other potential applications where temporal ordering is preserved?

**Questions:**

Please also see weakness.

1. To clarify what part was contributed in this paper and taken from[1] can the authors write a pseudo code and highlight the lines? For instance: side-by-side comparison of the original DEQ algorithm and the authors' modified version for equivariant networks.

2.  In Fig 2 why at 0th iteration(for test error) the models start at such different errors? Happens to be that the model starting with lower initial error saturates at lower overall error at end of epochs. I am unclear if this performance gain is due to less parameters (less overfit)in DEQ or due to the time correlation introduced in this paper? Could you please compare DEQ models with and without the temporal correlation initialization, while keeping the parameter count constant?

3. I suggest to add a table with number of parameters across all models.

references:

[1]https://arxiv.org/abs/1909.01377

---

> ### Author Response · Authors · 2024-11-19
>
> > Temporal correlation in molecular dynamics (MD) datasets has been used effectively to initialize fixed-point solvers. I like this idea, but it relies on an assumption of ordered data, which breaks down during phase transitions. For instance, in the simple case of water melting, the transformation from solid to liquid occurs abruptly, disrupting the expected order and causing this approach to fail at the transition point.
>
> We understand why the behavior of DEQs around phase transitions causes concerns; an abrupt change between frames would reduce the speedup. However, even though phase transitions are fast on a human time scale, they are still slow on an MD simulation time scale. For example, the referred melting of water happens at a time scale of $10^{-10}$ while typical integration time steps are on the order of $10^{-15}$ [1]. Therefore, two consecutive frames in an MD simulation of melting water are still very similar. Generally, the time step $\delta t$ must be chosen based on the fastest timescale in the system to obtain physical consistency. Since DEQuiformer only relies on two consecutive frames being similar, which the reduced time steps ensure, the assumptions are always satisfied under standard settings (no temporal coarse-graining or the like), even for phase transitions.
> We think we caused this confusion by saying we make use of “temporal correlation”, which sounds like we use temporal correlation over longer physical time horizons, when in reality DEQuiformer only relies on two consecutive frames being similar. We will change this in our manuscript to make clear that our assumption is continuity/smoothness only between successive time steps.
>
> [1] https://www.researchgate.net/figure/Typical-time-scales-for-structural-and-electronic-processes-in-solids_fig3_225412230
>
>
> &nbsp;
>
>
> > This approach also struggles with datasets that lack a clear temporal order, where entries are jumbled and any sense of sequence is lost (as noted by the authors for datasets like rMD17 and OC20). This limitation restricts the applicability of the method, impacting one of the core contributions of this work.
>
> The temporal correlation on rMD17 and OC20 is only lost because the dataset authors shuffled the data, but the underlying data inherently exhibits a smooth trajectory. In a real-world scenario, such shuffling would not occur, and this limitation would, therefore, not be a problem.
>
> To address this concern further, we ran an MD simulation of OC20 relaxations for 100-time steps and 100 starting points from the test set with and without fixed point reuse. Our results show that we can still use fixed point reuse, going from 29 to 11 iterations per step. We will include these results in the paper.
>
>
> &nbsp;
>
> > Can you please discuss potential ways to address this limitation or expand on other potential applications where temporal ordering is preserved?
>
> Temporal continuity is ubiquitous in all of material science and chemistry, from protein docking to combustion simulation to catalysis simulation and many more. The class of problems with temporal continuity is, therefore, enormous.
> Thinking outside the box, though, another potential application that might be possible would be lead optimization. In drug design or material science, we often have lead molecules and modify them by replacing singular small fragments while keeping the rest of the molecule the same. Since the overall structure doesn't change much, it might be possible to reuse the fixed point features of the atoms that are the same in both systems.
> Finally, there is this work  [2] that significantly speeds up DEQ convergence even in the absence of temporal continuity by predicting good starting points for the equilibration.
>
> [2] https://openreview.net/pdf?id=B0oHOwT5ENL

---

> ### Author Response · Authors · 2024-11-19
>
> > To clarify what part was contributed in this paper and taken from[1] can the authors write a pseudo code and highlight the lines? For instance: side-by-side comparison of the original DEQ algorithm and the authors' modified version for equivariant networks.
>
> Our DEQ takes building blocks from different published works and replaces the backbone layer with Equiformer. The original DEQ paper [3] used a linear initialization of the input injection, whereas we use Equiformer’s encoder. We also added a decoder (force and energy prediction heads). The solver is similar, but we use Anderson acceleration instead of Broyden’s method, and add a normalisation after each input injection. The original DEQ initialised fixed-points as zeros, whereas we took inspiration from [4] and initialized with the previous fixed-point. From [2] we also take the fixed-point correction loss and the relaxed solver tolerance.
> The main change we made to Equiformer was to remove alpha dropout as it hurt performance and replace path dropout with a recurrent path dropout. We will add pseudocode in the appendix describing the algorithm.
>
> [3] Deep Equilibrium Models https://arxiv.org/abs/1909.01377
>
> [4] Deep Equilibrium Optical Flow Estimation https://arxiv.org/abs/2204.08442
>
>
> &nbsp;
>
> > In Fig 2 why at 0th iteration(for test error) the models start at such different errors? Happens to be that the model starting with lower initial error saturates at lower overall error at end of epochs.
>
> This is because the first test evaluation happens after 50 training batches, which is enough for the models to show their performance differences. If you look in the appendix, in Figure 5 b), we plot the test loss for OC20 where the 4-layer EquiformerV2 actually starts at the same level as the DEQs. We also double-checked the validation loss (not in the paper), which starts being computed at the 0th batch, and all the models have very comparable errors.
>
> &nbsp;
>
> > I am unclear if this performance gain is due to less parameters (less overfit)in DEQ or due to the time correlation introduced in this paper?
>
> The time correlation is not the reason for the improved accuracy but used to speed up our model. You are right that DEQuiformer uses fewer parameters. Importantly though, these fewer parameters still result in a more expressive model due to the repeated application of the layers. If overfitting was the major factor, the 1-layer EquiformerV2 would perform similarly to the 1-layer DEQuiformer, but it is doing much worse.
> Another argument against overfitting in the traditional sense is that none of the test loss curves go up at any point of training. To reinforce this point, we trained the OC20 models 3 times longer, and the relative performance of all models in favor of DEQuiformer is the same, only with overall better errors for all models (we include this experiment in the appendix).
> Finally, as pointed out by one of the other reviewers, the ground truth DFT data are also the solution of a fixed point problem. Therefore, the increased performance might in part be due to an inductive bias.
>
> &nbsp;
>
> > Could you please compare DEQ models with and without the temporal correlation initialization, while keeping the parameter count constant?
>
> For speed, this is done in Figure 4 b, where we plot the average number of solver steps with and without reusing the fixed points for Aspiring. Reusing fixed points reduces the number of solver steps from ~5-6 to ~3 and, therefore, speeds up the model. As mentioned above, we also repeated the experiment for an OC20 MD simulation, where the speed-up is even more dramatic, from ~29 to ~11.
> For accuracy, we added the difference between the predictions with and without fixed point reuse to the appendix. The difference is below 1% on average for both Aspirin and the OC20-based relaxation simulations.
>
> &nbsp;
>
> > I suggest to add a table with number of parameters across all models.
>
> Thank you for the suggestion, we added a column with the number of model parameters to the tables.

---

> > ### Comment · Reviewer_yjG1 · 2024-11-26
> >
> > I appreciate the authors answering my questions promptly. I will stick to my current score of 6(marginally above the acceptance threshold). Interesting work!

---

> ### Author Response · Authors · 2024-11-28
>
> Thanks so much for your reply and engagement! We are glad to hear that our rebuttal answered all of your questions and that you find our work interesting. Please let us know if there are any other changes we can make.

---

### Official Review · Reviewer_iSQW · 2024-11-07

**Soundness:** 3
**Presentation:** 3
**Contribution:** 3
**Rating:** 5
**Confidence:** 3

**Summary:**

The authors follow techniques proposed in [1], [2], and [3] to convert EquiformerV2 into a Deep Equilibrium Model (DEQuiformer), leading to improvements in accuracy and inference speed on MD17, MD22, and OC20 200k datasets. The benefits of DEQuiformer are primarily derived from the ability to re-use previously inferenced fixed points as initializations for the next frame's inference.

**Strengths:**

- Can trade off accuracy for simulation speed post-training very easily by modulating the fixed-point error threshold on the solver
- The model outperforms EquiformerV2 on training time
- The model outperforms EquiformerV2 on MD17, MD22, and OC20 200K in accuracy and inference time
- The authors empirically show that DEQuiformer successfully and stably converges to a fixed point during inference

**Weaknesses:**

- Current work only applies to direct force prediction models
- No evidence that the technique could work on different model architectures
- No current comparisons with the inference times and accuracies of other model architectures

**Questions:**

Questions
- On line 278: "Using consecutive samples would have the downside of a large variance in the results depending on the starting index, while 1000 uniformly spaced samples yield similar results to expensively testing on all > 100,000 data points." I'm not sure if I understand this. Depending on the starting index, the accuracy will have high variance compared to the accuracy on testing on all 100,000 data points? Given a fixed datapoint, does this mean the accuracy of the model is different depending on that datapoint's position in the overall sampled trajectory? I'm not sure if this should be the case.

- Do shock simulations where consecutive frames lead to vastly different atomic arrangements lead to different performance/timing results? Are there guarantees that fixed-point reuse leads to less solver steps?


Limitations
- This is spoken in the limitations section, but I believe a very big drawback of the method is that it only applies to direct force prediction models. Real world molecular dynamics simulation necessitates forces to be formulated as gradients of energies to maintain important physical properties regarding the conservative nature of the force field. Although it's possible to run molecular dynamics simulation using a direct force prediction force field, it's still not clear whether this is trustworthy and safe to do so. Relaxations, on the other hand, don't as rigorously require conservative force fields but also don't require "millions to billions" of timesteps. Molecular dynamics is a fantastic beneficiary of the proposed speedup of inference, but the method isn't applicable at the moment.

---

> ### Author Response · Authors · 2024-11-19
>
> > “On line 278: "Using consecutive samples…." I'm not sure if I understand this. Depending on the starting index, the accuracy will have high variance compared to the accuracy on testing on all 100,000 data points? Given a fixed datapoint, does this mean the accuracy of the model is different depending on that datapoint's position in the overall sampled trajectory? I'm not sure if this should be the case.”
>
> Thank you for this important question. The observation is correct - the accuracy does depend on the sample's position in the trajectory, and this is true for both DEQuiformer and Equiformer. While this might seem counterintuitive, it reflects the inherently varying complexity of molecular configurations along the trajectory. For example, when Docosahexaenoic acid is uncurled, there are fewer complicated multi-body interactions compared to when it is curled up. Similarly, in AT-AT or AT-AT-CG-CG systems, fragments that are far separated have fewer interactions, making predictions relatively easier. Additionally, some sections of the trajectory naturally contain more states that are close to the densely sampled equilibrium geometries, placing them closer to the training distribution.
>
>
> &nbsp;
>
> > Do shock simulations where consecutive frames lead to vastly different atomic arrangements lead to different performance/timing results?
>
> Intuitively, it seems plausible that shock simulations lead to faster-moving atoms and, therefore, more dissimilar frames. However, to run a stable simulation, the time step $\delta t$ must be chosen based on the fastest timescale in the system to avoid dissipation, such that we always get smooth dynamics even for large shocks.
>
> Since DEQuiformer only relies on two consecutive frames being similar, the assumptions are always satisfied under standard settings (no temporal coarse-graining or the like). We think we caused this confusion by saying we make use of “temporal correlation”, which sounds like we use temporal correlation over longer physical time horizons when, in reality, DEQuiformer only relies on two consecutive simulation frames being similar. We will change this in our manuscript to make clear that our assumption is continuity/smoothness only between successive time steps.
>
> &nbsp;
>
> > Are there guarantees that fixed-point reuse leads to less solver steps?
>
> There are no mathematical guarantees, but empirically, it always holds true for our experiments. To strengthen this claim, we conducted additional experiments on OC20 MD simulations: We chose 100 random starting points from the test set and simulated the trajectory for 100 time steps, both with and without fixed point reuse. The average number of solver steps went from roughly 29 without to 11 with fixed point reuse, a saving of 62%.
>
> &nbsp;
>
>
> > No current comparisons with the inference times and accuracies of other model architectures
>
> The paper focuses on EquiformerV2 and DEQuiformer to ensure a fair, controlled comparison between these models. As our focus was on the relative performance gains from introducing the DEQ mechanism, we did not aim to provide a comprehensive benchmark against other existing models.

---

> > ### Comment · Reviewer_iSQW · 2024-11-27
> > **Thank you!**
> >
> > Thanks for the clarifications, I keep my score. Interesting work.

---

> ### Author Response · Authors · 2024-11-19
>
> > Current work only applies to direct force prediction models.
>
> > This is spoken in the limitations section, but I believe a very big drawback of the method is that it only applies to direct force prediction models. Real world molecular dynamics simulation necessitates forces to be formulated as gradients of energies to maintain important physical properties regarding the conservative nature of the force field.
>
> While we only show direct force prediction, one could simply use autograd to train and deploy DEQ-MLFF with energy-gradient force prediction, but one would lose the memory benefits during training.
> We have updated our explanation in the limitations.
>
>
> > Although it's possible to run molecular dynamics simulation using a direct force prediction force field, it's still not clear whether this is trustworthy and safe to do so. Relaxations, on the other hand, don't as rigorously require conservative force fields but also don't require "millions to billions" of timesteps. Molecular dynamics is a fantastic beneficiary of the proposed speedup of inference, but the method isn't applicable at the moment.
>
> We agree that MD simulations based on force prediction via the energy might be advantageous in some scenarios.
> However, we want to point out several other published works (besides Equiformerv2) that also use direct force prediction [1,2,3,4,5] and found important speed and numerical stability advantages. [4,5] both argued that direct force prediction works as well or better than energy-gradient-forces. Therefore, we think our work will still be of interest.
> Additionally, even though relaxations don't require millions of time steps per trajectory if used in a high throughput screening pipeline as intended for OC20, the model is still called millions of times during inference.
>
> [1] https://www.nature.com/articles/s41524-021-00543-3
>
> [2] https://www.nature.com/articles/s42256-019-0098-0
>
> [3] https://pubs.aip.org/aip/jcp/article/156/14/144103/2840972/Graph-neural-networks-accelerated-molecular
>
> [4] https://arxiv.org/pdf/2106.08903
>
> [5] https://arxiv.org/pdf/2204.02782
>
>
> &nbsp;
>
> > No evidence that the technique could work on different model architectures
>
> It would be very interesting to see if DEQs work for other model architectures as well. This would involve a significant increase in engineering and computing time and is outside the scope of our current work. Instead, we aim to demonstrate the potential of DEQs for one type of architecture. We expect the DEQ approach to be compatible with other MLFF architectures, since most build on a similar layer-wise message-passing scheme. Prior work also combined DEQs with a variety of other architectures and modalities[8], from convolutional neural networks[6], to transformers[7] to graph neural networks.
>
> [6] https://arxiv.org/pdf/2204.08442
>
> [7] https://arxiv.org/pdf/1909.01377
>
> [8] https://ieeexplore.ieee.org/abstract/document/10715398

---

> ### Author Response · Authors · 2024-11-28
>
> Thanks so much for your reply and engagement! We are glad to hear that our rebuttal answered all of your questions and that you find our work interesting. Please let us know if there are any other changes we can make.

---

### Author Response · Authors · 2024-11-19

We thank the reviewers for the constructive feedback and their questions. Some points were brought up by several reviewers, and we conducted additional experiments to address them:
1. We ran MD simulations on OC20 to measure the model’s solver step and runtime improvement due to fixed point reuse. DEQuiformer with 1-layer goes from 29 iterations without fixed point reuse to 11 with fixed point reuse while significantly beating a 14-layer EquiformerV2 in accuracy, the biggest we were able to fit in memory.
2. We measured the difference between force prediction with and without fixed point reuse on the full test set of Aspirin and OC20 MD simulations to show that the predictions are still Markovian: The average relative difference is less than 1% for both Aspirin, and the MD simulations.
3. We also trained the OC20 models three times longer and MD17 Aspirin twice as long and with up to ten times more parameters to address convergence concerns: While all models got better, the performance difference in favor of DEQuiformer remained.

We will add these experiments with detailed explanations to the paper.

---

### Meta-Review · Area_Chair_6uEe · 2024-12-22

**Metareview:**

In this work, authors propose DEQuiformer, which applies deep equilibrium models (DEQ) to machine learning force fields by modifying the EquiformerV2 architecture. The key innovation is leveraging temporal continuity in molecular dynamics (MD) simulations to accelerate convergence by reusing fixed points between timesteps. The work demonstrates that DEQuiformer achieves both higher accuracy and faster inference compared to EquiformerV2 on MD17, MD22, and OC20 datasets. Specifically, it shows 10x improvements while using fewer parameters. The authors validate that fixed point reuse reduces solver iterations from 29 to 11 steps on OC20 MD simulations while maintaining prediction accuracy within 1% relative difference.

The paper demonstrates several strengths as follows. The implementation achieves measurable improvements in both accuracy and computational efficiency for MD simulations, supported by comprehensive experiments added in response to reviewer feedback. The work integrates multiple existing techniques, including deep equilibrium models and optical flow estimation, while introducing necessary architectural modifications such as input injection normalization to optimize performance. A key innovation lies in leveraging temporal continuity in MD simulations, which Reviewer emZx particularly highlighted as having broader potential impact for modeling equilibrium properties in materials science. The authors' thorough response to reviewer concerns, including additional validation through MD simulations on OC20 and extensive convergence studies with scaled-up models and extended training periods, strengthens the paper's empirical foundation.

However, the work has several limitations that warrant consideration. As Reviewer edd9 points out, the current implementation is restricted to direct force prediction rather than energy gradient-based approaches, though the authors note this aligns with recent high-performing models in the field. However, it is well-known that this results in non-conservative forces. It raises questions about the stability of MD simulations and also the roughness of the underlying forcefield. More importantly, the consequence of the direct force prediction on basic physical laws such as energy and momentum conservation, an important aspect for MD simulations, remain questionable. While there are several direct force prediction models, most of them are applied for energy minimization or identifying stable structures rather than MD simulations. Additionally, the baseline comparisons rely on reduced datasets instead of full-scale training due to computational constraints, although scaling experiments indicate the relative performance improvements persist. The initial submission contained some inconsistencies in the MD22 evaluation results, though these were subsequently corrected during revision. These weaknesses, while notable, do not fundamentally undermine the paper's primary contributions and demonstrated practical utility.

Additional points are included below. Based on these points, the manuscript is not recommended for acceptance. For resubmission, the authors should focus on establishing theoretical foundations for DEQ in force fields, expanding to energy gradient-based approaches, and conducting thorough comparisons on standard benchmarks with complete training. This would help establish the approach's scientific merit beyond engineering optimization.

**Additional Comments On Reviewer Discussion:**

Some additional comments based on the reviewer discussions are as follows.

As emphasized by Reviewer edd9, the core contribution primarily consists of applying a 2019 DEQ technique to force fields without substantial theoretical advancement. The authors did not demonstrate fundamental insights about fixed point properties in molecular force fields or develop novel algorithmic improvements. The modifications made to the original DEQ approach, such as input injection normalization, represent engineering optimizations rather than conceptual advances.

Multiple reviewers identified significant concerns with the empirical evaluation. The energy prediction results on MD17 and MD22 are an order of magnitude worse than state-of-the-art, suggesting incomplete training or fundamental limitations. The authors acknowledge using reduced datasets and shorter training times due to computational constraints, making it difficult to validate claims about improvements over the full EquiformerV2 model. Reviewer edd9 notes that comparisons on incompletely trained models are not meaningful for establishing superiority.

The current approach only supports direct force prediction rather than energy gradient-based approaches, which limits its applicability for molecular dynamics simulations where conservative force fields are important. While the authors argue this is consistent with recent work, Reviewer iSQW emphasizes this as a significant drawback for real-world applications.

The initial experimental methodology had several issues requiring correction, including faulty numbers in the MD22 evaluation and inconsistent testing protocols compared to standard benchmarks. While some were addressed in revision, these raise concerns about the rigor of the empirical validation.

The authors' responses and additional experiments, while addressing some specific concerns, do not overcome the fundamental issues with innovation and comprehensive validation. The demonstrated improvements, while potentially useful in practice, do not represent sufficient scientific advancement for publication at a top-tier venue like ICLR. Future work would need to either develop more fundamental theoretical insights about fixed points in force fields or provide comprehensive validation on full-scale benchmarks with state-of-the-art comparisons.

---

### Decision · Program_Chairs · 2025-01-22

Reject